# A Multimodal LLM Approach for Visual Question Answering on Multiparametric 3D Brain MRI

## Abstract

We introduce mpLLM, a prompt-conditioned hierarchical mixture-of-experts (MoE) architecture for visual question answering over multiparametric 3D brain MRI (mpMRI). mpLLM routes across modality-level and token-level projection experts to fuse multiple interrelated 3D modalities, enabling efficient training without image–report pretraining. To address limited image-text paired supervision, mpLLM integrates a synthetic visual question answering (VQA) protocol that generates medically relevant VQA from segmentation annotations, and we collaborate with medical experts for clinical validation. mpLLM outperforms strong medical VLM baselines by 5.2% on average across multiple mpMRI datasets. Our study features three main contributions: (1) the first clinically validated VQA dataset for 3D brain mpMRI, (2) a novel multimodal LLM that handles multiple interrelated 3D modalities, and (3) strong empirical results that demonstrate the medical utility of our methodology. Ablations highlight the importance of modality-level and token-level experts and prompt-conditioned routing. We have included our source code in the supplementary materials and will release our dataset upon publication.

## 1 Introduction

Multiparametric MRI (mpMRI) plays a significant role in diagnosing, grading, treating, and assessing treatment responses for brain tumors and other intracranial lesions (Sawlani et al., 2020; Wang et al., 2022a; Cherubini et al., 2016). Describing imaging that involves a complex pattern of brain lesions across multiple regions can be challenging and time-consuming for clinicians. Consequently, several studies have been conducted to develop image recognition and localization models to support clinicians (Ghadimi et al., 2025; Rathore et al., 2018; Wang et al., 2022a; Li et al., 2023c; Osman, 2019).

However, existing models have limited clinical utility because clinicians cannot effectively pose natural language queries about mpMRI. While 3D vision-language models (VLMs) have been developed for other imaging domains, current architectures do not naturally leverage the interdependencies among mpMRI modalities (Li et al., 2023a; Wu et al., 2023; Bai et al., 2024; Xin et al., 2025). Additionally, the standard multi-image approach multiplies the number of vision tokens by the number of images, which significantly increases computational constraints (Wu et al., 2023).

We introduce mpLLM, a prompt-conditioned hierarchical mixture-of-experts (MoE) for VQA over mpMRI. Our approach is an extension of the LLaVa architecture Liu et al. (2023b), tailored to multiparametric MRI. In our approach, instead of the simple LLaVa-based projection function, we leverage a prompt-conditioned hierarchical MoE projection function, which generates a weighted average of high-level expert blocks to fuse the different sequences in multiparametric MRI for a more effective and efficient visual token representation for the LLM. Unlike modality-specific or modality-agnostic vision encoders (which must be trained independently and are difficult to train), we use lightweight projection functions that train end-to-end with the language model during fine-tuning.

To address limited image-text paired supervision, we pair mpLLM with a synthetic VQA protocol that derives medically relevant VQA from segmentation annotations, and we obtain clinician validation of both the generated data and model responses. In contrast to prior works, we fine-tune our model using next-token prediction directly on the VQA dataset without pretraining on a paired imaging-report

Figure 1: High-level comparison between LLaVA-Med and mpLLM. While LLaVA-Med uses a standard projection layer, our method uses a hierarchical MoE block which ingests both the prompt and imaging to produce prompt-conditioned vision tokens that leverage all the 3D modalities.

dataset. We also train a multi-task head end-to-end with the multimodal LLM for improved task proficiency and more reliable evaluation. In summary, our research makes these key contributions:

1. In collaboration with medical experts, we introduce a synthetic VQA protocol that produces the first clinically validated VQA dataset for 3D brain mpMRI.
2. We design mpLLM, a multimodal LLM that uses a prompt-conditioned hierarchical MoE to effectively leverage the interdependence between 3D modalities in mpMRI.
3. Strong empirical results that support our methodology as a foundation for future research with multimodal LLMs in brain mpMRI.

## 2 RELATED WORK

**Medical vision-language models**   Most vision-based medical multimodal LLMs can be broadly classified into CLIP-based discriminative models (Radford et al., 2021; Wang et al., 2022b; Eslami et al., 2023; Zhang et al., 2023a; Xu et al., 2024; Zhou et al., 2024; Huang et al., 2023) and LLM decoder-based generative models (Zhang et al., 2023b; Li et al., 2023a; Moor et al., 2023). Although discriminative models have proven helpful for various image recognition tasks, they possess limited utility in generation tasks such as VQA or report generation. Several popular generative models including MedVInt (Zhang et al., 2023b), LLaVA-Med (Li et al., 2023a), and MedFlamingo (Moor et al., 2023) share very similar architectures. However, these architectures and many others (Liu et al., 2024c; Lin et al., 2023; Li et al., 2023b; Zhu et al., 2024a; Lin et al., 2025; Zhang et al., 2025b; Nath et al., 2024; Guo et al., 2025) are designed specifically for 2D medical imaging and are not tailored to handle multiple 3D medical image modalities.

Although several 3D VLMs exist for natural images (Zhu et al., 2024b; Li et al., 2024b; Zhu et al., 2023), they require access to extremely large annotated datasets, which are often unavailable in medical contexts. While a few 3D VLMs have been developed for medical imaging, these methods have certain limitations. In one recent paper, researchers adapted the LLaVA-Med architecture to utilize spatial pooling and pretrain a 3D vision encoder with 700k radiology images (Bai et al., 2024). In another paper, researchers pretrain segmentation modules to generate brain imaging reports (Lei et al., 2024). In recent work, researchers exploit vision-language pretraining for CT report generation (Liu et al., 2023a; Chen & Hong, 2024; Blankemeier et al., 2024; Xin et al., 2025; Cao et al., 2025; Li et al., 2025a). However, these prior works assume a large paired imaging-report pretraining dataset, which is infeasible to collect and imposes a significant training burden.

Furthermore, previous methods focus on report generation instead of VQA, leading to less precise feedback regarding model strengths and weaknesses. Additionally, some models train directly on segmentation annotations (Lei et al., 2024; Rui et al., 2024), which are impractical to obtain, especially for novel use cases. Moreover, none of the previously discussed methods are tailored to handle multiple interdependent 3D image modalities, like in mpMRI, as input.

**Mixture-of-experts**   Previous work in MoE has concentrated on training and inference efficiency (Shazeer et al., 2017; Lepikhin et al., 2020; Fedus et al., 2022; Zoph et al., 2022; Liu et al., 2024a),

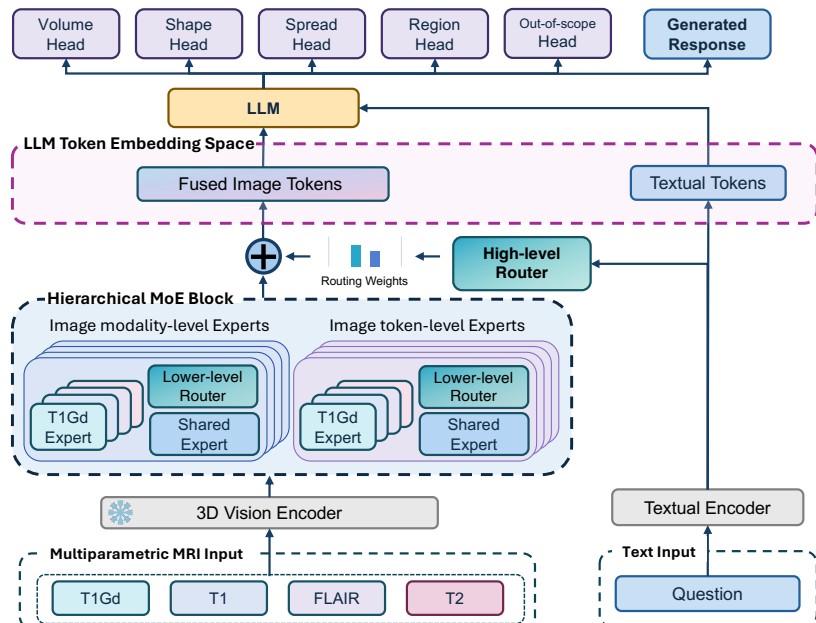

Figure 2: Detailed overview of our mpLLM pipeline.

transfer learning (Li et al., 2022; Zhong et al., 2022), class imbalance (Han et al., 2024), and multi-domain information (Zhang et al., 2024). There have also been earlier efforts with multimodal LLMs, covering sparsity learning (Lin et al., 2024), task interference (Shen et al., 2025), and embedding models (Li & Zhou, 2024). Related to our work, several studies have employed MoE with VLMs to select between vision encoders and vision-language projections (Li et al., 2025b; Zong et al., 2024; Wang et al., 2023; Ma et al., 2025). However, these studies address two modalities and do not account for interactions between different 3D image modalities, which present additional challenges our work seeks to address.

MoE also has various applications in the medical field. These applications include addressing missing modalities (Yun et al., 2025; Novosad et al., 2024; Liu et al., 2024d), fairness (Wang et al., 2025), pediatric care (Huy et al., 2025), parameter reduction and efficiency (Jiang et al., 2024; Nathani et al., 2024), and super resolution (Lin et al., 2021). Additionally, several studies have focused on the segmentation of multimodal medical imaging (Zhang et al., 2025a; Jiang & Shen, 2024). However, no existing research has explored using MoEs for multiple interrelated 3D image modalities. This area is particularly complex due to the need to project multiple interrelated vision modalities into the language modality.

**Medical VQA Datasets**    One of the primary challenges in report generation is evaluation: lexical metrics such as BLEU, ROUGE-L, and BERTScore have been shown to correlate poorly with radiologist evaluations (Yu et al., 2023). In contrast, VQA allows for more granular and interpretable model evaluation. While there are several medical VQA datasets, many focus on 2D imaging (Liu et al., 2024b; 2021; He et al., 2020; Lau et al., 2018). In a prior work, researchers used a scene graph generator to generate surgical VQA (Yuan et al., 2024). In a recent work, researchers extracted multi-task questions from structured lung cancer screening data (Niu et al., 2025). However, there is no existing VQA dataset for 3D brain mpMRI due to a significant lack of source data for VQA extraction. In our work, we remedy this by leveraging publicly available segmentation annotations as source data.

## 3 METHODOLOGY

Brain mpMRI has several 3D imaging modalities. For a given modality $m$, let $I_m \in \mathbb{R}^{C \times D \times H \times W}$ denote the corresponding 3D volume, where $C$, $D$, $H$, and $W$ represent the channel, depth, height, and width, respectively. A 3D vision encoder $h_{\text{vis}}$ maps each modality to a sequence of image tokens $v_m = h_{\text{vis}}(I_m) \in \mathbb{R}^{T \times d_V}$, where $T$ is the number of image tokens and $d_V$ is the vision embedding

dimension. We apply spatial pooling to reduce the token length (reusing the symbol $T$ for the pooled length for simplicity) and then concatenate the pooled tokens across the $M$ image modalities before passing them to the hierarchical MoE.

Let $v \in \mathbb{R}^{M \times T \times d_V}$ denote the concatenated image modality embeddings. The hierarchical MoE projects these embeddings into the LLM space, $e = \text{MoE}(v, t) \in \mathbb{R}^{M \times T \times d_T}$, where $\text{MoE}$ denotes the hierarchical MoE block, $t$ represents the text prompt, and $d_T$ is the LLM embedding dimension. In practice we flatten the modality and token dimensions and provide $(t, e)$ as a soft prompt to the LLM for multi-task prediction and text generation. A detailed visualization of our approach can be seen in Figure 2.

## 3.1 HIERARCHICAL MIXTURE-OF-EXPERTS FOR MULTIPARAMETRIC MRI PROJECTION

In what follows, we use the term *expert* exclusively for projection modules that map vision features into the LLM embedding space, and the term *router* for the MLPs that output mixing weights over experts. Each high-level *expert block* therefore consists of a router together with its associated projection experts.

**High-level router**    Our hierarchical MoE architecture includes a high-level router $r^{(h)}$ that assigns weights over a set of high-level expert blocks $\{\mathcal{E}_1^{(h)}, \ldots, \mathcal{E}_H^{(h)}\}$, where $H$ is the number of high-level experts. These expert blocks operate at the image modality and image token levels. The router is implemented as a two-layer MLP that takes as input the final hidden state of the language model corresponding to the text prompt $t$. It produces a normalized weight distribution over expert blocks: $\pi^{(h)}(t) = \text{softmax}(r^{(h)}(t)) \in \mathbb{R}^H$. Since task information is embedded within the text prompt, the router implicitly infers the task, enabling high-level experts to specialize in different task proficiencies.

**High-level image modality-level and image token-level experts**    Our hierarchical MoE includes high-level experts operating at different granularity levels: image modality-level and image token-level. Each high-level expert block consists of a two-layer MLP low-level router $r^{(l)}$ and an associated set of low-level projection experts $\{W_1^{(l)}, \ldots, W_L^{(l)}\}$, where $L$ is the number of low-level experts within the block.

The image modality-level expert block uses a low-level router that takes as input the concatenated [CLS] tokens from all image modalities (e.g., T1, T2) and outputs modality-level weights over the corresponding low-level experts: $\pi_{\text{mod}}^{(l)}(v) = \sigma(r_{\text{mod}}^{(l)}(v)) \in \mathbb{R}^L$. In contrast, the token-level expert block uses a low-level router that receives, for each token position $i$, the $i$-th image tokens from all modalities and outputs token-level weights $\pi_{\text{tok}}^{(l)}(v) = \sigma(r_{\text{tok}}^{(l)}(v)) \in \mathbb{R}^{L \times T}$. As discussed in prior work (Li et al., 2024a), providing weights at different granularities improves task performance by enhancing domain generalizability.

**Low-level image modality-specific and image modality-agnostic (shared) experts**    Each low-level expert $W$ represents a projection transformation from the vision encoder embedding space to the LLM embedding space: $W : R^{N_I \times d_I} \rightarrow R^{N_I \times d_T}$. We utilized a simple linear transformation for the projection transformation as in the original LLaVa paper (Liu et al., 2023b). Each image modality embedding is processed through a modality-specific expert and a modality-agnostic (shared) expert. The modality-specific expert emphasizes extracting image modality-specific features, whereas the modality-agnostic expert focuses on deriving common features from all image modalities. The parameters for the modality-specific expert are unique to each image modality (T1Gd, T1, T2, and FLAIR), while those for the modality-agnostic expert are consistent across all image modalities.

Each image modality is passed through both low-level experts and then summed embedding dimension-wise. The overall formulation for the hierarchical MoE is as follows:

$$\text{MoE}(v, t) = \sum_{h=1}^{H} \pi_h^{(h)}(t) \sum_{m=1}^{M} \left( \alpha_m^{(h)} W_m^{(h)}(v_m) + \beta_m^{(h)} W_{\text{shared}}^{(h)}(v_m) \right), \tag{1}$$

Table 1: Statistics for the synthetic VQA datasets.

| Dataset | # questions | # mpMRI | # unique questions | # unique answers |
|---------|-------------|---------|--------------------|--------------------|
| GLI | 38,904 | 1,621 | 38,023 | 36,773 |
| MET | 11,718 | 651 | 11,607 | 11,284 |
| GoAT | 24,318 | 1,351 | 23,859 | 23,223 |

where $h$ indexes the high-level expert blocks, $m$ indexes the image modalities, and $\alpha_m^{(h)}$ and $\beta_m^{(h)}$ are the modality-specific and modality-agnostic weights produced by the corresponding low-level router (which sum to one within an expert block $h$). Our validation experiments on the GLI dataset found that the optimal number of high-level experts was 16, corresponding to the number of labels times the number of tasks.

The fused image token embeddings are combined with the text prompt token embeddings. Then, these embeddings are input into the LLM decoder at the token embedding layer for multi-task prediction and text generation.

## 3.2 TRAINING OBJECTIVES

### 3.2.1 SYNTHETIC VQA PROTOCOL

Because of the lack of brain mpMRI VQA data, we propose a novel method of synthetic VQA generation that leverages the publicly available brain mpMRI segmentation data. To generate relevant VQA data, we consult with clinicians to identify important topics that can be extracted from the label masks, focusing on mask volume relative to brain volume (Kaifi, 2023), brain region localization (Lau et al., 2018), shape (Ismail et al., 2018), and spread (Islam et al., 2019). For each label mask, we compute the quantities using standard formulas and validate the thresholds with synthetic masks and a subset of data. To emulate the subjectivity found in medical reports, we categorize each of the quantities based on their magnitude using terminology similar to that found in medical reports. Rather than using an LLM, we employ a rules-based method to assign medical terms to the quantities, ensuring our approach is clinically relevant and highly reliable. We assign "N/A" if the label is not found.

**Volume**  To calculate the relative mask volume, we determine the number of mask pixels and divide by the number of brain pixels in the volume (which are the nonzero pixels in the skull-stripped T1 image modality). The subjective labels we use are "$< 1\%$", "$1 - 5\%$", "$5 - 10\%$", "$10 - 25\%$", "$25 - 50\%$", and "$50 - 75\%$".

**Region**  For the BraTS GLI volumes, we use the Nibabel python library (Abraham et al., 2014) to register the volumes to the AAL atlas (version SPM12) Rolls et al. (2020) and extract the following brain regions: "frontal", "parietal", "occipital", "temporal", "limbic", "insula", "subcortical", and "cerebellum". For the BraTS MET and GoAT volumes, we register the volumes to the LPBA40 atlas (in SRI24 space) (Shattuck et al., 2008) and extract the following brain regions: "frontal", "parietal", "occipital", "temporal", "limbic", "insula", "subcortical", "cerebellum", and "brainstem". The percent coverages of the segmentation masks with the atlases are 67.3%, 70.9%, and 57.7% for the GLI, MET, and GoAT datasets respectively.

**Shape**  We first quantify each mask's overall size and compute classical 3-D shape metrics (sphericity, elongation, flatness, solidity, compactness). If the mask is tiny, it is classified as "focus"; otherwise, we classify it as "round," "oval," "elongated," or "irregular" by comparing its sphericity and elongation values to empirically chosen thresholds that correspond to near-sphere, mildly flattened, and strongly stretched geometries.

**Spread**  We identify all disconnected islands, noting the largest as the "core," and compute what proportion of the total mask volume it occupies. If there is only one island, the pattern is "single lesion"; if multiple islands are present but the core retains $\geq 70\%$ of the volume, it is described as

"core with satellite lesions"; otherwise, when no dominant island exists, the distribution is marked "scattered lesions."

**Question-answer pair generation**  After computing the previous quantities for each label mask, we create a dataset that simulates the natural variability of human input. First, we consider all combinations of the four major tasks to create multi-task question-answer pairs. After we have the 15 question-answer pair types, we use ChatGPT-4o to generate approximately 3000 perturbations of each question-answer pair (without affecting the label and answer term) that emulates the language a clinician would use. We also add question-answer pairs with partially out-of-scope and completely out-of-scope tasks to improve the model's self-awareness of its capabilities. Thus, for each label and mpMRI in each dataset, we sample four multitask question-answer pairs without replacement such that each major task is addressed in at least one question-answer pair, one partially out-of-scope question-answer pair, and one completely out-of-scope question-answer pair. Examples of generated question-answer pairs can be seen in the Appendix in Table 6. The answers are used as supervision for next-token prediction for the multimodal LLM.

### 3.2.2 MULTI-TASK HEADS

For increased task proficiency and more accurate task evaluation, we train a multi-task head end-to-end with the multimodal LLM. After providing the soft-prompt to the multimodal LLM, we extract the hidden state from the last layer and apply task-specific heads (which consist of a single linear layer) to generate multi-task predictions. For volume, shape, spread, and out-of-scope task identification, the task is multi-class classification, and the associated loss is categorical cross-entropy; whereas for region localization, the task is multi-label classification, and the associated loss is multi-label binary cross-entropy. These losses are added to the next-token prediction loss to produce our multi-task loss:

$$\mathcal{L} = \mathcal{L}_{\text{Next-token}} + \mathcal{L}_{\text{Volume}} + \mathcal{L}_{\text{Region}} + \mathcal{L}_{\text{Shape}} + \mathcal{L}_{\text{Spread}} + \mathcal{L}_{\text{Out-of-scope}} \quad (2)$$

## 4 EXPERIMENTS

### 4.1 DATASETS DETAILS

For our synthetic VQA protocol, we leverage the 2024 Brain Tumor Segmentation (BraTS) challenge (LaBella et al., 2024), which provides a standardized benchmarking environment for automated brain tumor segmentation. All datasets comprise of co-registered multiparametric MRI scans (T1, T1Gd, T2, FLAIR) at $1\text{mm}^3$ resolution, skull-stripped and manually annotated by experts. To enable fair comparison and manage GPU memory, all BraTS sequences were resampled to $32 \times 256 \times 256$. This allowed for compatibility with baseline methods, such as M3D (Bai et al., 2024) and Med3DVLM (Xin et al., 2025). We consider three challenges in BraTS: GLI, MET and GoAT. The challenges are collected from over ten institutions and encompass diverse pathological contexts and imaging protocols.

**GLI (Adult Glioma Post Treatment)** focuses on post-treatment diffuse glioma segmentation and consists of multi-institutional routine post-treatment clinically-acquired multiparametric mpMRI scans of glioma. The task requires the delineation of enhancing tumor (ET), non-enhancing tumor core (NETC), surrounding FLAIR hyperintensity (SNFH), and resection cavity (RC) (de Verdier et al., 2024).

**MET (Brain Metastases)** contains a retrospective compilation of treatment-naive brain metastases mpMRI scans obtained from various institutions under standard clinical conditions. The challenge addresses the segmentation of small metastatic lesions using a 3-label system (NETC, SNFH, ET) and demonstrates variable tumor component distribution across cases (Moawad et al., 2024).

**GoAT (Generalizability Across Tumors)** assesses algorithmic generalizability across different tumor types (i.e., different number of lesions per scan, lesion sizes, and locations in the brain), institutions (i.e., different MRI scanners, acquisition protocols), and demographics (i.e., different age, sex, etc.). The challenge uses consistent labels (necrosis, edema/invaded tissue, and enhancing tumor) despite varying tumor morphology to evaluate algorithm adaptability to new disease types with

Table 2: Comparison of task performance for all models on all datasets with accuracy metric with standard deviation.

| Dataset | Method | Volume | Region | Shape | Spread | Mean |
|---|---|---|---|---|---|---|
| GLI | RadFM (Wu et al., 2023) | 13.1±0.8 | 68.5±0.5 | 17.6±0.9 | 17.0±0.9 | 29.0±0.4 |
| | Med3DVLM (Xin et al., 2025) | 31.3±1.1 | 72.9±0.4 | 42.0±1.2 | 37.5±1.2 | 45.9±0.6 |
| | M3D (Bai et al., 2024) | 39.7±1.2 | 73.4±0.5 | 53.7±1.1 | 52.9±1.2 | 54.9±0.6 |
| | LLaVA-Med (Li et al., 2023a) | 40.2±1.2 | 76.3±0.5 | 52.1±1.2 | 49.7±1.3 | 54.6±0.7 |
| | mpLLM (Ours) | **62.0±1.2** | **83.0±0.4** | **57.6±1.2** | **57.2±1.2** | **64.9±0.6** |
| MET | RadFM | 12.8±1.3 | 69.5±1.0 | 13.8±1.5 | 13.3±1.3 | 27.3±0.7 |
| | Med3DVLM | 44.4±1.9 | 70.1±1.0 | 34.3±1.9 | 35.2±1.8 | 46.0±1.0 |
| | M3D | **66.7±1.9** | 73.3±0.9 | 50.9±1.9 | 43.4±1.9 | 58.6±1.0 |
| | LLaVA-Med | 45.9±2.1 | 68.5±1.0 | 38.9±2.0 | 34.7±1.9 | 47.0±1.0 |
| | mpLLM (Ours) | 65.8±2.0 | **76.4±0.8** | **52.9±2.0** | **52.7±2.0** | **62.0±1.0** |
| GoAT | RadFM | 11.9±1.0 | 64.7±0.6 | 35.1±1.4 | 28.1±1.3 | 34.9±0.6 |
| | Med3DVLM | 33.0±1.5 | 65.4±0.6 | 58.4±1.4 | 51.2±1.4 | 52.0±0.7 |
| | M3D | 57.6±1.5 | 75.4±0.6 | 76.0±1.2 | 65.5±1.4 | 68.6±0.7 |
| | LLaVA-Med | 59.4±1.4 | 76.7±0.5 | **76.4±1.2** | 67.2±1.3 | 69.9±0.6 |
| | mpLLM (Ours) | **64.4±1.4** | **77.2±0.5** | 73.4±1.3 | **67.8±1.3** | **70.7±0.7** |

limited training data (de Verdier et al., 2024; Moawad et al., 2024; LaBella et al., 2023; Kazerooni et al., 2024; Adewole et al., 2023).

To generate the train, validation, and test sets, we randomly sample 80%, 10%, and 10% from the imaging studies. For GLI we generated 31,104, 4,176, and 3,624 question-answer pairs for the train, validation, and test sets based on 1,621 mpMRIs. For MET, we generated 9,090, 1,368, and 1,260 question-answer pairs for the train, validation, and test sets, based on 651 mpMRIs. For GoAT, we generated 19,440, 2,430, and 2,448 question-answer pairs for the train, validation, and test sets, based on 1351 mpMRIs.

**Clinical validation**

We collaborated with two radiologists who annotated 20 mpMRIs from the BraTS-GLI test set, 10 mpMRIs from the BraTS-MET test set, and 10 mpMRIs from the BraTS-GoAT test set with questions spanning four tasks and four findings for the BraTS-GLI dataset and three findings for the BraTS-MET and BraTS-GoAT datasets, yielding a total of 560 questions. We used 10 annotated mpMRIs from BraTS-GLI for validation to improve the task label thresholds for the synthetic data. We used the other 30 annotated mpMRIs to evaluate the interannotator agreement between the synthetic data and the radiologist, obtaining 55.1% accuracy and compared this with the agreement between the two radiologists, obtaining 58.9% accuracy (see Table 10). We observe that the accuracies are very similar, indicating that the synthetic data quality is comparable to radiologist-annotated data. The primary reason that the synthetic data agreement accuracy is reduced relative to the second annotator is due to the region accuracy, which is primarily dependent on the quality of the brain-atlas registration.

To assess the quality of our synthetic questions, a radiologist evaluated the clarity of 160 synthetic questions from 10 mpMRIs from the BraTS-GLI test set using a binary scoring system (1 = valid, 0 = invalid). The synthetic questions achieved a 92.2% validity rate, indicating high acceptability. More statistics about the synthetic datasets can be seen in Table 1, and more details can be found in Appendix A and B.

## 4.2 EXPERIMENTAL SETTINGS

**Models** In our experiments, we utilized the Phi-3-Mini-4K-Instruct LLM. We also explored utilizing the Llama models and chose Phi-3 because of the increased efficiency and negligible performance benefits of the Llama models. For versatility and generality, we utilize the 3D Vision Transformer (3D ViT) (Dosovitskiy et al., 2020) as the vision encoder and use medically pretrained weights (Bai et al., 2024).

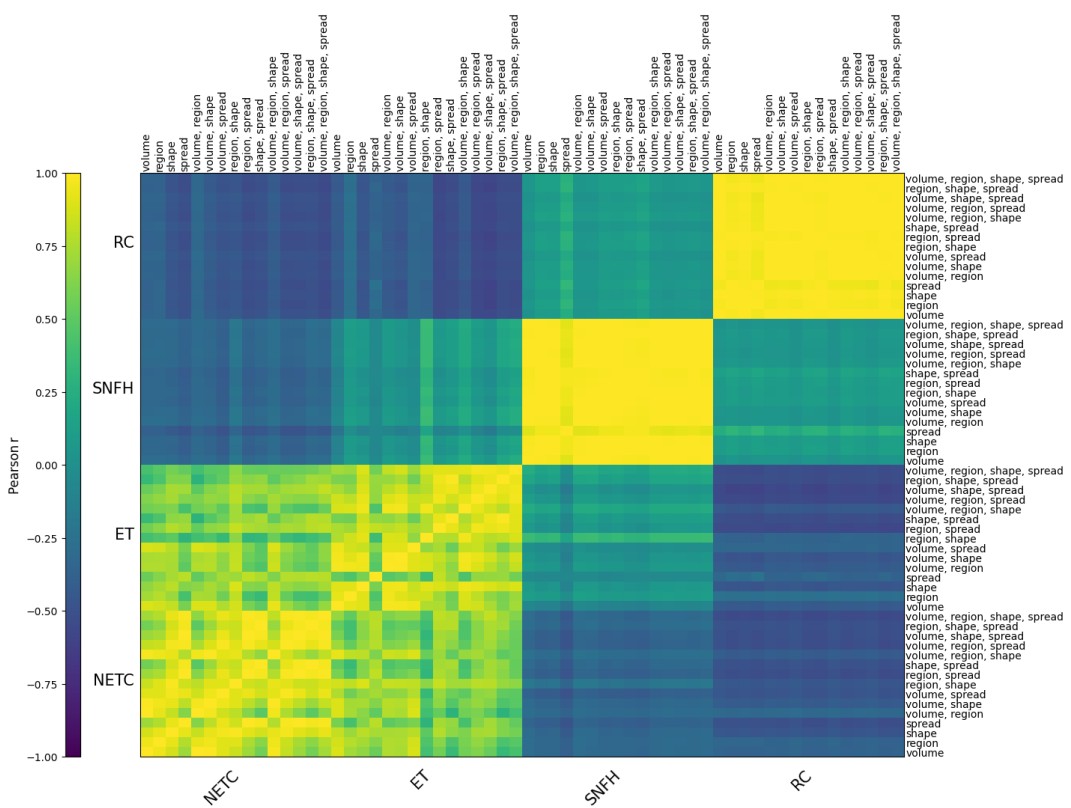

Figure 3: Heatmap for correlation between high-level expert weight vectors for standard prompts in the GLI dataset. NETC = non-enhancing tumor core, ET = enhancing tissue, SNFH = surrounding FLAIR hyperintensity, RC = resection cavity.

**Training** We fine-tune the multimodal LLM using the loss defined in Equation 2 on the VQA training dataset. We freeze the vision encoder while unfreezing the hierarchical MoE and LLM weights. We train the model on the train dataset for 2 epochs. The LLM is trained with LoRA, setting $r$ to 16 and $alpha$ to 32, with a dropout of 0.1. We employ a cosine learning rate scheduler that starts at a learning rate of $2.0 \times 10^{-4}$.

**Baseline models** We compare our approach to several baseline models, including LLaVA-Med (Li et al., 2023a), M3D (Bai et al., 2024), Med3DVLM (Xin et al., 2025), and RadFM (Wu et al., 2023)[1]. To process the multiple 3D MRI image modalities, we use a multi-image approach, in which we concatenate the image tokens generated from each MRI image modality from a shared projection layer and vision encoder (Wu et al., 2023). Because LLaVA-Med is not implemented with a 3D vision encoder, to ensure a fair comparison, we test it with our model's vision encoder (Bai et al., 2024). Similar to our method, the vision encoder is frozen and only the projection layer and LLM are trainable. To provide a comparison to our model's multi-task heads, which are trained end-to-end with the rest of our framework, we independently train a new multi-task head. We use a Phi3 language model with multi-task heads to predict the multi-task outputs given the prompt and text generation. The model is trained on our train dataset and had 99.8% accuracy on the validation set. Other hyperparameter settings mirror our method as closely as possible to ensure a fair comparison.

**Evaluation** For evaluating the models' task proficiency, we use accuracy for volume, shape, spread, and out-of-scope tasks, and per-label accuracy for the region task. We estimate the standard deviation using 500 bootstrap resamples.

---

[1]We planned to evaluate Merlin (Blankemeier et al., 2024), but the report-generation model weights were not publicly available at the time of submission.

Table 3: Ablation study on the MoE architecture on the GLI validation set with accuracy metric.

| Modality-level MoE | Token-level MoE | Prompt-based MoE weights | Task Mean |
|---|---|---|---|
| ✗ | ✗ | ✗ | 63.3 |
| ✓ | ✗ | ✗ | 64.1 |
| ✗ | ✓ | ✗ | 64.4 |
| ✓ | ✓ | ✓ | **65.5** |

**Computing environment**    All our experiments were mainly conducted using a single NVIDIA A100 GPU on an internal cluster. Training our model on the GLI dataset took roughly 8 hours.

Table 4: Radiologist acceptance rate comparison between mpLLM and M3D.

| Model | Radiologist Acceptance Rate (%) |
|---|---|
| M3D | 34.1 |
| mpLLM | **50.0** |

Table 5: Comparison of model performance for differentiating primary gliomas versus secondary metastatic lesions

| Model | Accuracy | AUROC |
|---|---|---|
| M3D | 88.5 | 95.5 |
| mpLLM | **95.6** | **99.0** |

## 4.3 RESULTS

All model results across the evaluated datasets are presented in Table 2. Our model consistently achieves strong performance across all task categories and datasets, outperforming the second-best model by an average margin of 5.2%. Furthermore, it ranks first in nearly all sub-categories and datasets, highlighting both its broad capabilities and strong generalizability.

Examining the memory usage, our model only required approximately 20 GB of GPU memory during training and inference – significantly less than M3D, LLaVA-Med, and Med3DVLM, all of which exceed 40 GB – suggesting the computational benefits of a fused vision token representation. In our experiments, we also noticed that the top three models had above a 99.8% accuracy on out-of-scope task identification, which suggests our dataset was effective at hallucination mitigation.

## 4.4 ABLATION STUDIES

The ablation study on the MoE architecture is in Table 3. Image modality-level and token-level high-level MoE experts perform better than the single projection layer baseline approach. A prompt-conditioned weighted combination of the different high-level experts performs the best.

Fine-grained results comparing our MoE-based approach and a single shared expert are in Table 14 and Table 15. The more complex multimodal reasoning is helpful for all tasks and findings, and especially helpful for the volume task and SNFH finding. Fine-grained results comparing modality-level MoE, token-level MoE, as well prompt-conditioned MoE on tasks as well as findings are in Table 16 and Table 17. Token-level MoE is stronger in the volume, region, and spread tasks while modality-level MoE is stronger in the shape task. For findings, token-level MoE is stronger with the ET, SNFH, and RC findings while modality-level MoE is stronger with the NETC finding. The prompt-conditioned MoE excels in all of them, suggesting that based on the question, it is able to accurately combine the optimal token-level MoE or modality-level MoE blocks.

To qualitatively evaluate our architecture, we construct all 60 template task prompts from our GLI dataset (four findings $\times$ 15 task combinations = 60 template prompts) and input them into our model's

high-level router to generate high-level expert weight vectors. We then calculate the correlation between these weight vectors and generate a heatmap, which is in Figure 3. There's high correlation between expert weight vectors within the same finding, suggesting similar image features are extracted. For findings that are closer anatomically, such as non-enhancing tumor core and enhancing tissue, there is also relatively high correlation between the expert weight vectors. This is reasonable because of their close proximity anatomically, which suggests similar extracted image features. For findings like resection cavity and surrounding FLAIR hyperintensity that are more diverse anatomically from the other findings, there's much lower correlation, which again is sensible.

## 4.5 CLINICAL UTILITY

In order to validate the usefulness of the model generated responses, we collaborated with a radiologist to conduct a user study. We created a clinical validation set of 208 questions stemming from 13 cases from the GLI test set, each question focusing on either volume, region, shape, or spread (specifically questions 1 through 4 as described in Appendix A). We had mpLLM as well as M3D (one of the most competitive baseline models) provide responses to these questions and we asked the clinician to independently evaluate each model's response as sufficient or lacking. The results are in Table 4. While there is still more work to be done before clinical deployment, the results are quite promising. Additionally, we see a significant margin between our approach and the current baseline.

In order to further emphasize the importance of the model extracted features from brain mpMRI, we have constructed an additional downstream task, which aims to differentiate primary gliomas from secondary metastatic lesions based on imaging patterns, using features like volume, region, shape, and spread. For this task, we combined the BraTS GLI and MET datasets, which have samples with primary gliomas and secondary metastatic lesions respectively. Because our models are trained with guard rails and indicate if tasks are outside of the scope of what they were trained on, we fitted logistic regression models based on the model generated features on the train set to predict primary glioma versus secondary metastatic lesions and evaluated them on the test set, which can be seen in Table 5. mpLLM additionally performs well on this important clinical task, scoring 7 percentage points and 4 percentage points higher than the M3D model on the accuracy and AUROC metrics respectively.

## 5 CONCLUSION

We present mpLLM, a multimodal LLM with prompt-conditioned hierarchical MoE that routes across modality- and token-level projection experts for mpMRI VQA, enabling efficient end-to-end fine-tuning without paired image–report pretraining. With a clinician-validated synthetic VQA pipeline derived from segmentation annotations, mpLLM improves over strong medical VLM baselines by an average of +5.2% while using <50% GPU memory. Ablations highlight the modality/token experts, prompt-conditioned routing, and an integrated multi-task head. Strong results on user studies as well as downstream tasks suggest high potential for clinical use. Future work includes open-ended VQA/report generation, broader multi-reader validation, and fairness analyses.

## 6 ETHICS STATEMENT

This work uses publicly available, fully de-identified BraTS datasets, minimizing risks to patient privacy and data security. Our synthetic VQA questions are generated from segmentation annotations, and both the generated questions and model outputs underwent clinician review to mitigate typical risks of synthetic supervision. Nonetheless, fairness and bias remain open concerns: synthetic prompts and limited demographic metadata can yield models that underperform for underrepresented groups or clinical scenarios. The model is intended for research only and must not be used for autonomous clinical decision-making; it is designed to abstain on out-of-scope queries, and any deployment would require prospective, multi-site validation under qualified clinical oversight. In future work, we will expand evaluations to demographically diverse cohorts where available, document dataset composition and known limitations, and incorporate explicit fairness analyses and bias-mitigation strategies alongside robustness and calibration assessments.

# 7 REPRODUCIBILITY STATEMENT

We use a publicly available dataset and detail the full data-generation pipeline in Sections 3.2.1 and 4.1, with additional information in Appendix A. We include documented code in the supplementary materials, and report all experimental settings and computational resources in Section 4.2.

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

# A   ADDITIONAL INFORMATION REGARDING DATASET

In the following, we will describe the formulas used to derive the shape and spread descriptors for our synthetic VQA protocol. Let $M \subset \mathbb{Z}^3$ be a binary mask of foreground voxels sampled with spacing $\mathbf{s} = (s_x, s_y, s_z)\,[\text{mm}]$ (typically $s_x = s_y = s_z = 1$). Write $\Delta V = s_x s_y s_z$ for the physical volume of one voxel and $|M|$ for the number of foreground voxels.

**Total volume**
$$V_{\text{tot}} = |M|\,\Delta V \;[\text{mm}^3].$$

**Multiplicity**   We decompose $M$ into 26-connected components $M_1, \ldots, M_{N_c}$ (scipy 'ndimage.label' with a unit "ball" structuring element) and record $N_c$.

**Spread**   Let the *core component* index be $i^\star = \arg\max_i V_i$. Define

$$f_{\text{core}} \;=\; \frac{V_{i^\star}}{V_{\text{tot}}} \in [0, 1].$$

$$\text{spread} = \begin{cases} \text{``single lesion''} & N_c = 1, \\ \text{``core with satellite lesions''} & N_c > 1, \; f_{\text{core}} \geq 0.7, \\ \text{``scattered lesions''} & \text{otherwise.} \end{cases}$$

For each component $M_i$:

**Component surface area**   Marching cubes (scikit-image 'measure.marching_cubes') produces a triangular mesh $(\mathcal{V}_i, \mathcal{F}_i)$ in real-world coordinates. The mesh area (which we describe as the surface area) is

$$A_i = \sum_{(p,q,r) \in \mathcal{F}_i} \tfrac{1}{2} \big\| (q - p) \times (r - p) \big\|_2.$$

**Component volume**   $V_i = |M_i|\,\Delta V$.

**Component sphericity**
$$\Phi_i \;=\; \frac{\pi^{1/3}\,(6V_i)^{2/3}}{A_i}.$$

**Component compactness**
$$C_i \;=\; \frac{A_i}{V_i}.$$

**Component principal-axis statistics**   Assemble voxel coordinates $\mathbf{x}_j = (x_j, y_j, z_j) \in \mathbb{R}^3$ for $j \in M_i$. The covariance matrix $\Sigma_i = \frac{1}{|M_i|} \sum_j (\mathbf{x}_j - \bar{\mathbf{x}})(\mathbf{x}_j - \bar{\mathbf{x}})^\top$ yields eigenvalues $\lambda_1 \geq \lambda_2 \geq \lambda_3 > 0$.

**Component elongation**
$$E_i \;=\; \sqrt{\lambda_1/\lambda_2}.$$

**Component flatness**
$$F_i \;=\; \sqrt{\lambda_3/\lambda_2}.$$

**Component solidity**   A convex hull (scipy 'ConvexHull') provides volume $V_i^{\text{hull}}$;

$$S_i \;=\; \frac{V_i}{V_i^{\text{hull}}}.$$

**Metric aggregation**

$$(\Phi, E, F, S, C) = \begin{cases} (\Phi_{i^\star}, E_{i^\star}, F_{i^\star}, S_{i^\star}, C_{i^\star}) & N_c = 1 \text{ or } f_{\text{core}} \geq 0.7, \\ \dfrac{1}{N_c} \sum_{i=1}^{N_c} (\Phi_i, E_i, F_i, S_i, C_i) & \text{otherwise.} \end{cases}$$

**Shape**  Convert the continuous metrics to one of five categories:

$$\text{shape} = \begin{cases} \text{"focus"} & V_{\text{tot}} < 0.1 \text{ cm}^3 \quad (\ V_{\text{tot}} \times 10^{-3} < 0.1\ ) \\ \text{"round"} & \Phi \geq 0.85 \ \wedge\ E \leq 1.3, \\ \text{"oval"} & 0.60 \leq \Phi < 0.85 \ \wedge\ 1.3 < E \leq 2.5, \\ \text{"elongated"} & E > 2.5, \\ \text{"irregular"} & \text{otherwise.} \end{cases}$$

The thresholds were set empirically on a development set of annotated masks and match clinicians' qualitative intuition of near-spherical, mildly flattened, and strongly stretched geometries. All computations are implemented in Python using scipy, scikit-image, numpy, and ndimage as shown in the listing above.

**Question augmentation details**  We use ChatGPT to generate question augmentations of our multitask dataset. For generating question augmentations for the standard multi-task prompts, we first provide this prompt "Please produce hundred alternative wordings that a clinician may use for the following question and answer. Please include everything surrounded by curly braces {} as they are because they are placeholders. Please generate the reworded question starting with "Q:" and reworded answer starting with "A:" and separate each generated question-answer pair with a newline. Please do not produce any additional text." and append this to each of the multitask prompts below. We produce 40 repetitions with a temperature of 1.0, top p of 1, and model "gpt-4o-mini-2024-07-18".

1. Q: How large is the volume covered by {label}? A: The overall volume of {label} is {volume}.

2. Q: Which region(s) of the brain is {label} located in? A: The {label} is located in {regions}.

3. Q: What is the shape of {label}? A: The shape of {label} is {shape}.

4. Q: How spread out is {label}? A: The spread of {label} is {spread}.

5. Q: How large is the volume of {label} and where is it located? A: The overall volume of {label} is {volume}, and it is located in {regions}.

6. Q: How large is the volume of {label} and what is its shape? A: The overall volume of {label} is {volume}, and its shape is described as {shape}.

7. Q: How large is the volume of {label} and how spread out is it? A: The overall volume of {label} is {volume}, and it is characterized as {spread}.

8. Q: In which region is {label} and what is its shape? A: The {label} is located in {regions}, and its shape is described as {shape}.

9. Q: In which region is {label} and how spread out is it? A: The {label} is located in {regions}, and it is characterized as {spread}.

10. Q: What is the shape of {label} and how spread out is it? A: The shape of {label} is described as {shape}, and it is characterized as {spread}.

11. Q: What is the volume, region, and shape of {label}? A: The overall volume of {label} is {volume}, it is located in {regions}, and its shape is described as {shape}.

12. Q: What is the volume, region, and spread of {label}? A: The overall volume of {label} is {volume}, it is located in {regions}, and it is characterized as {spread}.

13. Q: What is the volume, shape, and spread of {label}? A: The overall volume of {label} is {volume}, its shape is described as {shape}, and it is characterized as {spread}.

Table 6: Qualitative examples illustrating the four question types generated by our pipeline.

| Type | Question | Answer |
|---|---|---|
| Single-task | Could you identify the shape of the Resection Cavity? | The shape of the Resection Cavity is **irregular**. |
| Multi-task | How do you quantify the volume of Surrounding Non-enhancing FLAIR hyperintensity, and what does its distribution look like? | The total volume of Surrounding Non-enhancing FLAIR hyperintensity is described as **1–5%**, characterized as **core with satellite lesions**. |
| Partial-out-of-scope | What areas does Surrounding Non-enhancing FLAIR hyperintensity encompass, what is its volume, and synthesize findings into a comprehensive care plan? | Surrounding Non-enhancing FLAIR hyperintensity has a volume of **1–5%** in **cerebellum, frontal and parietal**. Synthesizing this into a care plan is *outside my domain*. |
| Out-of-scope | How do genetic factors influence the development of Non-Enhancing Tumor in adolescents? | *I cannot provide information* regarding the influence of genetic factors on the development of Non-Enhancing Tumor in adolescents. |

14. Q: What is the region, shape, and spread of {label}? A: The {label} is located in {regions}, its shape is described as {shape}, and it is characterized as {spread}.

15. Q: What is the volume, region, shape, and spread of {label}? A: The overall volume of {label} is {volume}, it is located in {regions}, its shape is described as {shape}, and it is characterized as {spread}.

For generating question augmentations for the partially out-of-scope multi-task prompts, we first provide this prompt "Please produce hundred alternative wordings that a clinician may use for the following question and answer and incorporate an additional clinical task or tasks which the model cannot solve in the reworded question. These can be before, after, or interspersed between the other tasks (please make sure to vary the order and number of out-of-scope tasks). Do not mention that the model cannot answer these in the question; however, indicate that the model cannot answer that part of the question in the reworded answer (potentially using different phrasings). The model can describe the volume, brain region, shape, and spread of {label} which is the region of interest. Please include everything surrounded by curly braces {} as they are because they are placeholders. Please generate the reworded question starting with "Q:" and reworded answer starting with "A:" and do not produce any additional text." and append this to each of the multitask prompts above. We produce 10 repetitions with a temperature of 1.0, top p of 1, and model "gpt-4o-2024-08-06".

For generating question augmentations for completely out-of-scope prompts, we first provide this prompt "Please produce a hundred questions (with one or more tasks) that a clinician may ask that the model does not have information to answer. The model can describe the volume, brain region, shape, and spread of {label} which is the region of interest. Please include {label} in the question but do not include anything else with curly braces. In the answer, please indicate the model cannot answer the question (potentially using different phrasings). Please generate the question starting with "Q:" and answer starting with "A:" and do not produce any additional text." We produce 10 repetitions with a temperature of 1.0, top p of 1, and model "gpt-4o-mini-2024-07-18".

After generating the question augmentations, we check the generated results for quality (ensuring the contents within the curly braces are retained for easy formatting with Python and that the responses are in English). Then, for each finding and mpMRI in each dataset, we sample four multitask questions without replacement such that each major task is addressed in at least one question, one partially out-of-scope question, and one completely out-of-scope question. Examples of generated question types can be seen in Table 6.

After the application of our synthetic VQA protocol, the percentage frequency of each task per question for all the generated datasets can be seen in Table 7, Table 8, and Table 9.

Table 7: Percentage frequency of each task label per question for the GLI dataset

| Task | Label name | Label frequency |
|------|-----------|-----------------|
| Volume | Unspecified | 52.4 |
| | N/A | 12.3 |
| | <1% | 12.7 |
| | 1-5% | 13.1 |
| | 5-10% | 5.2 |
| | 10-25% | 3.8 |
| | 25-50% | 0.5 |
| | 50-75% | 0.0 |
| Region | Unspecified | 53.2 |
| | N/A | 12.2 |
| | frontal | 23.8 |
| | parietal | 20.8 |
| | occipital | 13.2 |
| | temporal | 17.2 |
| | limbic | 21.7 |
| | insula | 14.7 |
| | subcortical | 14.9 |
| | cerebellum | 2.9 |
| | Unspecified | 52.4 |
| | N/A | 12.3 |
| | focus | 1.0 |
| | round | 4.7 |
| | oval | 6.9 |
| | elongated | 0.4 |
| | irregular | 22.4 |
| Spread | Unspecified | 53.4 |
| | N/A | 12.2 |
| | single lesion | 6.9 |
| | core with satellite lesions | 20.9 |
| | scattered lesions | 6.5 |
| Out-of-scope | Not out-of-scope | 66.7 |
| | Out-of-scope | 33.3 |

## B    ADDITIONAL ABLATION RESULTS

Additional ablation results validating the number of high-level experts, softmax versus sigmoid for summing lower-level experts, and concatenation versus element-wise summing of vision tokens are in Table 11, Table 12, and Table 13 respectively. In Table 18, we see a comparison of our model performance trained with our multi-task loss versus the next-token prediction baseline loss. There is a significant performance improvement with our multi-task loss.

Fine-grained results comparing our MoE-based approach and a single shared expert are in Table 14 and Table 15. The more complex multimodal reasoning is helpful for all tasks and findings, and especially helpful for the volume task and SNFH finding. Fine-grained results comparing modality-level MoE, token-level MoE, as well prompt-conditioned MoE on tasks as well as findings are in Table 16 and Table 17. Token-level MoE is stronger in the volume, region, and spread tasks while modality-level MoE is stronger in the shape task. For findings, token-level MoE is stronger with the ET, SNFH, and RC findings while modality-level MoE is stronger with the NETC finding. The

Table 8: Percentage frequency of each task label per question for the MET dataset

| Task | Label name | Label frequency |
|------|-----------|-----------------|
| Volume | Unspecified | 51.8 |
| | N/A | 8.5 |
| | <1% | 24.7 |
| | 1-5% | 9.2 |
| | 5-10% | 2.7 |
| | 10-25% | 2.7 |
| | 25-50% | 0.4 |
| | 50-75% | 0.0 |
| Region | Unspecified | 53.0 |
| | N/A | 17.1 |
| | frontal | 19.5 |
| | parietal | 16.1 |
| | occipital | 14.5 |
| | temporal | 14.5 |
| | limbic | 9.2 |
| | insula | 6.3 |
| | subcortical | 7.3 |
| | cerebellum | 12.8 |
| | brainstem | 4.1 |
| Shape | Unspecified | 51.4 |
| | N/A | 8.3 |
| | focus | 2.8 |
| | round | 13.5 |
| | oval | 4.3 |
| | elongated | 0.2 |
| | irregular | 19.5 |
| Spread | Unspecified | 53.1 |
| | N/A | 7.9 |
| | single lesion | 7.4 |
| | core with satellite lesions | 12.9 |
| | scattered lesions | 18.7 |
| Out-of-scope | Not out-of-scope | 66.7 |
| | Out-of-scope | 33.3 |

prompt-conditioned MoE excels in all of them, suggesting that based on the question, it is able to accurately combine the optimal token-level MoE or modality-level MoE blocks.

We also constructed an additional experiment in which, for each finding, we appended the region information to the prompt, which can be seen in Table 19. There is an increase of approximately 5% on the other task scores with the additional localization prompt. This suggests that providing the localization information can be extremely beneficial.

We also evaluated the frequency of hallucination. We define a hallucination as the instance in which a model predicts a non-zero finding volume and the ground-truth indicates there is no finding. Additionally, we define a correct prediction as the instance in which the model predicts a non-zero finding volume and the ground-truth also indicates a non-zero finding volume. We report the percent of hallucinations over total predictions (correct predictions + hallucinations) based on finding on the GLI test set in Table 20. While this is out of scope for our current work, it is notable that the model is already achieving no hallucinations for SNFH.

Table 9: Percentage frequency of each task label per question for the GoAT dataset

| Task | Label name | Label frequency |
|---|---|---|
| Volume | Unspecified | 52.2 |
| | N/A | 2.2 |
| | <1% | 10.3 |
| | 1-5% | 18.6 |
| | 5-10% | 8.9 |
| | 10-25% | 7.2 |
| | 25-50% | 0.6 |
| Region | Unspecified | 52.9 |
| | N/A | 2.3 |
| | frontal | 30.0 |
| | parietal | 23.4 |
| | occipital | 17.5 |
| | temporal | 29.7 |
| | limbic | 29.8 |
| | insula | 25.9 |
| | subcortical | 27.9 |
| | cerebellum | 9.8 |
| | brainstem | 8.0 |
| Shape | Unspecified | 51.9 |
| | N/A | 2.1 |
| | focus | 0.5 |
| | round | 6.1 |
| | oval | 3.2 |
| | elongated | 0.1 |
| | irregular | 36.0 |
| Spread | Unspecified | 53.3 |
| | N/A | 1.9 |
| | single lesion | 5.3 |
| | core with satellite lesions | 32.7 |
| | scattered lesions | 6.7 |
| Out-of-scope | Not out-of-scope | 66.7 |
| | Out-of-scope | 33.3 |

Table 10: Agreement with first annotator

| Annotator | Multi-class Accuracy | Region Accuracy | Task Mean Accuracy |
|---|---|---|---|
| second annotator | 50.0 | 74.2 | 58.9 |
| synthetic groundtruth | 48.7 | 85.5 | 55.1 |

## C  LLM USAGE

We used large language models (LLMs) to (i) improve the clarity and style of the manuscript, (ii) brainstorm refinements to the MoE-based architecture and dataset-construction procedures, (iii) draft code prototypes for selected ideas, and (iv) find potentially relevant related work. All LLM outputs were reviewed and verified by the authors before inclusion.

Table 11: Model performance comparison with different number of high-level experts on the GLI validation set with accuracy metric.

| Number of blocks | Task Mean |
| --- | --- |
| 12 | 64.5 |
| 16 | **65.5** |
| 20 | 65.0 |

Table 12: Model performance with softmax versus sigmoid for summing lower-level experts on the GLI validation set with accuracy metric.

| Method | Task Mean |
| --- | --- |
| softmax | **65.5** |
| sigmoid | 64.8 |

Table 13: Comparison of projection and fusion methods on the GLI validation set with accuracy metric.

| Projection Method | Fusion Method | Task Mean |
| --- | --- | --- |
| MoE-based | sum | **65.5** |
| Shared expert | learned weighted sum | 63.3 |
| Shared expert | sum | 51.1 |
| Shared expert | concatenation | 52.4 |

Table 14: Comparison of MoE-based and shared expert approach based on task on the GLI validation set with accuracy metric

| Method | Volume | Region | Shape | Spread |
| --- | --- | --- | --- | --- |
| MoE-based | **57.6** | **83.5** | **60.9** | **59.9** |
| Shared expert | 53.2 | 82.8 | 59.7 | 57.5 |

Table 15: Comparison of MoE-based and shared expert approach based on finding on the GLI validation set with accuracy metric

| Method | ET | SNFH | NETC | RC |
| --- | --- | --- | --- | --- |
| MoE-based | **65.1** | **72.5** | **75.1** | **49.9** |
| Shared expert | 63.2 | 70.8 | 71.3 | 48.7 |

Table 16: Comparison of MoE approaches based on task on the GLI validation set with accuracy metric

| Method | Volume | Region | Shape | Spread |
| --- | --- | --- | --- | --- |
| Prompt-conditioned MoE | **57.6** | **83.5** | **60.9** | **59.9** |
| Token-level MoE | 56.0 | **83.5** | 60.0 | 58.1 |
| Modality-level MoE | 53.7 | 83.4 | 59.7 | 59.4 |

Table 17: Comparison of MoE approaches based on finding on the GLI validation set with accuracy metric

| Method | ET | SNFH | NETC | RC |
| --- | --- | --- | --- | --- |
| Prompt-conditioned MoE | **65.1** | **72.5** | **75.1** | **49.9** |
| Token-level MoE | 64.0 | 72.2 | 72.2 | **49.9** |
| Modality-level MoE | 63.5 | 71.1 | 73.9 | 48.6 |

Table 18: Model comparison with multi-task loss on the GLI validation set with accuracy metric.

| Method | Task Mean |
|--------|-----------|
| mpLLM without multi-task loss | 56.7 |
| mpLLM with multi-task loss | **65.5** |

Table 19: Comparison of task means with and without region information on the GLI validation set with accuracy metric.

| Model | Task Mean without Region Scores |
|-------|--------------------------------|
| With Region Information | **64.1** |
| Without Region Information | 59.5 |

Table 20: Hallucination frequency across findings on the GLI test set.

| | ET | SNFH | NETC | RC |
|--------|------|------|------|------|
| Values | 22.6 | 0.0 | 13.5 | 15.3 |

