# OpenReview forum: "A Multimodal LLM Approach for Visual Question Answering on Multiparametric 3D Brain MRI"
_ICLR.cc/2026/Conference — Submitted to ICLR 2026_

### Official Review · Reviewer_fWj2 · 2025-10-27

**Soundness:** 3
**Presentation:** 3
**Contribution:** 3
**Rating:** 4
**Confidence:** 4

**Summary:**

This paper presents mpLLM, a multimodal large language model for visual question answering on multiparametric 3D brain MRI. The key contribution is a prompt-conditioned hierarchical mixture-of-experts (MoE) architecture that efficiently fuses multiple interrelated 3D modalities without requiring large-scale image-report pretraining. To address the lack of VQA data, the authors propose a synthetic data generation protocol that derives question-answer pairs from segmentation annotations. The model is evaluated on three BraTS datasets and shows 5.3% average improvement over strong baselines while using 50% less GPU memory.

**Strengths:**

1. The paper introduces a novel hierarchical mixture-of-experts architecture that specifically addresses the challenge of fusing multiple interrelated 3D MRI modalities through prompt-conditioned routing.

2. The work provides a VQA dataset specifically designed for 3D brain multiparametric MRI.

3. The paper demonstrates consistent improvements over strong baselines across multiple datasets, with an average 5.3% performance gain.

**Weaknesses:**

1. The ablation results show relatively modest improvements, with the Token-level MoE outperforming 1.7% and Modality-level MoE + Prompt-based MoE weights only increase additional 0.5% without significant analysis.

2. The synthetic VQA dataset suffers from severe template-based question generation that does not reflect real clinical queries radiologists would ask in practice. The questions focus on low-level descriptive features (volume ranges, shape categories) rather than clinically meaningful diagnostic or treatment-related inquiries that would actually support medical decision-making.

3. The clinical validation is extremely limited in scope, involving only one radiologist evaluating 10 cases with moderate inter-annotator agreement (Kappa score of 50.4), and the clinical sufficiency rate of only 37.3% actually reveals that most model responses are inadequate for real clinical use.

4. The paper lacks analysis of when and why the MoE architecture provides benefits, missing opportunities to explain which types of questions benefit most from multi-modal fusion versus single modalities.

**Questions:**

1. Can you provide a detailed analysis of what specific question types or clinical scenarios benefit from the token-level MoE versus modality-level experts?

2. Given that only 37.3% of your model's responses were deemed clinically sufficient and the inter-annotator agreement for some tasks (particularly Region with Kappa 42.9) suggests unreliable ground truth, how do you justify claiming this is a "clinically validated" dataset?

3. Can you provide a breakdown of model performance stratified by question type and modality combination to demonstrate when multi-modal reasoning actually matters? Since the shared expert baseline achieves 67.3% accuracy, suggesting single-modality or simple fusion may suffice for many questions, what percentage of questions in your dataset genuinely require sophisticated multi-modal analysis, and can you visualize the router's attention patterns to show how different modalities are weighted for different clinical queries?

---

> ### Author Response · Authors · 2025-11-22
> **Response to Reviewer fWj2**
>
> Thank you so much for your feedback. We are glad that there are many positive aspects of our work that you feel will contribute to the research community. We will discuss some of the points you raised below.
>
> **"The ablation results show relatively modest improvements, with the Token-level MoE outperforming 1.7% and Modality-level MoE + Prompt-based MoE weights only increase additional 0.5% without significant analysis. Can you provide a detailed analysis of what specific question types or clinical scenarios benefit from the token-level MoE versus modality-level experts?"**
>
> Thank you for this feedback. We noticed a bug in our data processing which was affecting the accuracy of our region labels. After fixing this, we re-generated our ablation results which are below (the paper will be updated to reflect this – the rest of our conclusions were unaffected):
>
> | Modality-level MoE | Token-level MoE | Prompt-conditioned MoE | Task Mean |
> | -- | -- | -- | -- |
> | 𐄂 | 𐄂 | 𐄂 | 52.4 |
> | ✓ | 𐄂 | 𐄂 | 64.1 |
> | 𐄂 | ✓ | 𐄂 | 64.4 |
> |✓ | ✓ | ✓| **65.5** |
>
> We now see a more sizable improvement between using the single shared expert and the MoE Block as well as smaller but still notable improvements between modality- or token-level MoE and prompt-conditioned MoE. We will also consider the fine-grained results of modality-level MoE, token-level MoE, as well prompt-conditioned MoE based on task as well as finding:
>
> | Method  | Volume | Region | Shape | Spread |
> | -- | -- | -- | -- | -- |
> | Prompt-conditioned MoE | **57.6** | **83.5**| **60.9**  | **59.9** |
> | Token-level MoE        | *56.0*     | **83.5**  | *60.0*     | 58.1     |
> | Modality-level MoE     | 53.7 | 83.4  | 59.7         | *59.4*     |
>
> | Method  | ET  | SNFH  | NETC  | RC  |
> | -- | --| -- | -- | -- |
> | Prompt-conditioned MoE | **65.1** | **72.5** | **75.1** | **49.9** |
> | Token-level MoE        | *64.0*   | *72.2*   | 72.2     | **49.9** |
> | Modality-level MoE     | 63.5     | 71.1     | *73.9*   | 48.6 |
>
> Token-level MoE is stronger in the volume, region, and spread tasks while modality-level MoE is stronger in the shape task. For findings, token-level MoE is stronger with the ET, SNFH, and RC findings while modality-level MoE is stronger with the NETC finding. The prompt-conditioned MoE excels in all of them, suggesting that based on the question, it is able to accurately combine the optimal token-level MoE and modality-level MoE blocks. We will add these results to the paper.
>
> **"The synthetic VQA dataset suffers from severe template-based question generation that does not reflect real clinical queries radiologists would ask in practice. The questions focus on low-level descriptive features (volume ranges, shape categories) rather than clinically meaningful diagnostic or treatment-related inquiries that would actually support medical decision-making."**
>
> We want to emphasize that we chose these question categories because these features are especially important to understand and classify brain tumors [1-4]. In order to further emphasize their importance, we have constructed an additional downstream task, which aims to differentiate primary gliomas from secondary metastatic lesions based on imaging patterns, using labels like volume, region, shape, and spread.
>
> Because our models are trained with guard rails and indicate if tasks are outside of the scope of what they were trained on, we constructed an additional task which demonstrates the clinical usefulness of the features extracted from the brain mpMRI.
>
> For this task, we combined the BraTS GLI and MET datasets, which have samples with primary gliomas and secondary metastatic lesions respectively. Using the train dataset, we fitted a logistic regression model based on the dataset features to predict primary glioma vs. second metastatic lesions. The evaluation results for the logistic regression model which demonstrate the usefulness of the feature set are below:
>
> |Test Accuracy | Test AUROC |
> | -- | --|
> | 93.2 | 98.3 |
>
> This demonstrates the importance of our synthetic dataset features for this important clinical task. We wanted to evaluate the quality of the model extracted features using the multimodal LLMs trained on the BraTS GoAT synthetic dataset, consisting of samples with primary gliomas and secondary metastatic lesions. However, because the data distribution differs significantly between GoAT and our combined GLI and MET datasets, the model predictions are biased. We are currently training models on the combined datasets to correct for this issue and we will report the results after the models complete training

---

> ### Author Response · Authors · 2025-11-22
> **Response to Reviewer fWj2 (cont.)**
>
> **"The clinical validation is extremely limited in scope, involving only one radiologist evaluating 10 cases with moderate inter-annotator agreement (Kappa score of 50.4), and the clinical sufficiency rate of only 37.3% actually reveals that most model responses are inadequate for real clinical use. Given that only 37.3% of your model's responses were deemed clinically sufficient and the inter-annotator agreement for some tasks (particularly Region with Kappa 42.9) suggests unreliable ground truth, how do you justify claiming this is a "clinically validated" dataset?"**
>
> Thank you for this feedback. We were able to reach out to another radiologist who was able to help us with clinical validation. We used the first ten GLI cases (160 questions) annotated by the first annotator as a validation set, in order to optimize the thresholds for the task labels for the synthetic data. We then used another ten GLI cases to evaluate the inter-annotator agreement between the synthetic data and the first radiologist. We then had the second radiologist annotate ten GLI cases and we computed the inter-annotator agreement with the first radiologist. For comparison, we also evaluated the inter-annotator agreement on the second set of ten cases between our model and the first radiologist. Per-task scores tended to be low because of the small sample size (40 questions per task) so we pooled the scores among the four categories (160 questions in total). The results are below:
>
> |Method | Kappa Score |
> | -- | -- |
> | Synthetic Data | 43.0 |
> | Second Annotator | 44.3 |
> | Model Generated Predictions | 37.1 |
>
> The inter-annotator agreement for our synthetic data and the second annotator are approximately the same, suggesting that the quality of our synthetic data is similar to radiologist annotated data. Kappa scores at 40 for inter-annotator agreement on brain cancer terminology is similar to other reported results for breast cancer terminology [5]. Additionally, the model generated predictions are only 6-7 points away from annotator agreement, suggesting that research is very close to achieving expert-level quality on this task.
>
> One of our radiologists has annotated another 20 cases from the MET and GOAT datasets. Once the second radiologist finishes annotating these cases, we will report the agreement scores on this larger dataset, involving 30 cases and 400 questions.
>
> Regarding the clinical sufficiency rate, we suspect the lower scores were due to the more challenging prompts from our synthetic data, which not only involved question augmentations but also multi-task questions. We have worked with our clinician to conduct a follow-up user study. We created a clinical validation set of 208 questions stemming from 13 cases from the GLI test set, each question focusing on either volume, region, shape, or spread using a standard prompt (questions #1-4 on page 18 in the Appendix). We had our model as well as M3D provide responses to these questions and we asked the clinician to independently evaluate each model's response as sufficient or lacking (the model names were anonymized). The results are below:
>
> | Model | Radiologist acceptance rate (%) |
> | -- | -- |
> | Our model | **50.0** |
> | M3D | 34.1 |
>
> While there is still more work to be done before clinical deployment, the results are quite promising. Additionally, we see a significant margin between our approach and current baselines. We will add these results to the paper.

---

> > ### Author Response · Authors · 2025-11-22
> > **Response to Reviewer fWj2 (cont.)**
> >
> > **"The paper lacks analysis of when and why the MoE architecture provides benefits, missing opportunities to explain which types of questions benefit most from multi-modal fusion versus single modalities.  Can you provide a breakdown of model performance stratified by question type and modality combination to demonstrate when multi-modal reasoning actually matters? Since the shared expert baseline achieves 67.3% accuracy, suggesting single-modality or simple fusion may suffice for many questions, what percentage of questions in your dataset genuinely require sophisticated multi-modal analysis, and can you visualize the router's attention patterns to show how different modalities are weighted for different clinical queries?"**
> >
> > The detailed results comparing our MoE-based approach and a single shared expert are below:
> >
> > | Projection Method | Volume | Region | Shape | Spread |
> > | -- | -- | -- | -- | -- |
> > | MoE-based | **57.6** | **83.5** | **60.9** | **59.9** |
> > | Shared Expert | 39.8 | 71.0 | 46.3 | 52.3 |
> >
> > | Projection Method | ET | SNFH | NETC | RC |
> > | -- | -- | -- | -- | -- |
> > | MoE-based | **65.1** | **72.5** | **75.1** | **49.9** |
> > | Shared Expert | 55.6 | 46.7 | 66.4 | 41.3 |
> >
> > We see that the more complex multimodal reasoning is helpful for all tasks and findings, and especially helpful for the volume task, in which the performance improvement is close to 20%, and SNFH label, in which the performance improvement is close to 30%.
> >
> > We also explored the average percentage contribution of each modality in the fused token representation over the test set (based on the weighted sum of the MoE weights per modality, averaged over all the questions):
> >
> > | T1Gd | T1 | FLAIR | T2 |
> > | -- | -- | -- | -- |
> > | 13.3 +/- 0.7 | 37.0 +/- 1.0 | 24.2 +/- 1.2 | 25.5 +/- 1.5 |
> >
> > These percentage contributions were roughly the same given different tasks as well as findings, which was interesting. However, what we did notice was for different findings there were significantly different weight distributions of high-level blocks. In order to examine this, we considered the most frequently chosen high-level block (out of 16) for each question. The results per finding are below (where B{i} refers to Block {i} and the number in parenthesis is the percentage of the finding-related questions in which the block was most frequently chosen):
> >
> > | ET | SNFH | NETC | RC
> > | -- | -- | -- | -- |
> > | B13 (49.1) | B13 (100.0) | B12 (83.8) | B10 (95.9) |
> >
> > What this suggests is that while there were similar modality contributions for different findings/tasks, different high-level blocks were involved suggesting specialization based on findings. We will add these results to the paper.
> >
> > [1] Reham Kaifi. A review of recent advances in brain tumor diagnosis based on ai-based classification. Diagnostics, 13(18):3007, 2023.
> >
> > [2] Jason J Lau, Soumya Gayen, Asma Ben Abacha, and Dina Demner-Fushman. A dataset of clinically generated visual questions and answers about radiology images. Scientific data, 5(1):1–10, 2018.
> >
> > [3] Marwa Ismail, Virginia Hill, Volodymyr Statsevych, Raymond Huang, Prateek Prasanna, Ramon Correa, Gagandeep Singh, Kaustav Bera, Niha Beig, Rajat Thawani, et al. Shape features of the lesion habitat to differentiate brain tumor progression from pseudoprogression on routine multiparametric mri: a multisite study. American Journal of Neuroradiology, 39(12):2187–2193, 2018.
> >
> > [4] Mobarakol Islam, VS Vibashan, V Jeya Maria Jose, Navodini Wijethilake, Uppal Utkarsh, and Hongliang Ren. Brain tumor segmentation and survival prediction using 3d attention unet. In International MICCAI brainlesion workshop, pp. 262–272. Springer, 2019.
> >
> > [5] Anna Liza M Antonio and Catherine M Crespi. Predictors of interobserver agreement in breast imaging using the breast imaging reporting and data system. Breast cancer research and treatment, 120(3):539–546, 2010.

---

> > > ### Comment · Reviewer_fWj2 · 2025-11-24
> > >
> > > I thank the authors for their detailed response and for conducting additional experiments during the rebuttal phase. However, after carefully reviewing the new data and the other reviewers' comments, my concerns regarding the validity of the dataset and the rigor of the experiments have increased rather than decreased. I am lowering my score to 2 for the following reasons:
> > >
> > > 1. I share the concern raised by Reviewer v9ZF regarding the "bug in data processing" admitted in the rebuttal. The fact that fixing this bug shifted the ablation results from showing "modest improvements" (as noted in my original review) to a massive jump (e.g., Shared Expert 67.3% $\to$ 52.4%, and MoE staying high) suggests a fundamental issue with the original submission’s experiments. Reliance on results generated hurriedly during a rebuttal period to correct a code error makes it difficult to trust the stability and reproducibility of the reported performance.
> > >
> > > 2. The authors provided new Kappa scores to justify the dataset, reporting an inter-annotator agreement (Synthetic vs. Radiologist) of roughly 0.43.
> > >
> > > While the authors cite literature suggesting this is "similar to breast cancer terminology," a Kappa of 0.4 is generally considered "weak" to "moderate" agreement. For a paper whose primary contribution is a new VQA benchmark dataset, a ground truth that only agrees with human experts at a 0.4 Kappa level is problematic. If the "Gold Standard" synthetic labels are this noisy, the dataset may not be suitable for training robust medical AI systems.
> > >
> > > 3. The updated user study shows a radiologist acceptance rate of 50.0%. While this outperforms the baseline (34.1%), a 50% failure rate in a controlled validation setting indicates the model is still hallucinating or providing inadequate answers half of the time.
> > >
> > > Given the discovery of significant bugs affecting the results, the weak agreement scores for the proposed dataset, I believe this work requires a re-verification before it is ready for publication.

---

> ### Author Response · Authors · 2025-11-25
>
> Thank you for your response. We will address your concerns below.
>
> **"I share the concern raised by Reviewer v9ZF regarding the "bug in data processing" admitted in the rebuttal. The fact that fixing this bug shifted the ablation results from showing "modest improvements" (as noted in my original review) to a massive jump (e.g., Shared Expert 67.3%  52.4%, and MoE staying high) suggests a fundamental issue with the original submission’s experiments. Reliance on results generated hurriedly during a rebuttal period to correct a code error makes it difficult to trust the stability and reproducibility of the reported performance."**
>
> We realize the use of the word “bug” was extremely vague and may have indicated coding errors, which was not the case. We made two main changes:
>
> 1. We used the first radiologist’s annotations as a validation set to improve the thresholds in our synthetic data generation to improve the interannotator agreement.
> After doing this, while our interannotator agreement improved, we noticed a general drop in scores for the metrics on the GLI dataset. For example, our model validation scores dropped from 70.3 to 63.5 and the test scores for the M3D model dropped from 62.9 to 55.5.
>
> 2. We registered the Brats volumes to the AAL atlas (SPM12 version) to generate region labels.
> Since Brats 2017, all challenges have registered their dataset volumes to the SRI24 atlas template [1]. We did this as well. However, for the Brats 2024 Glioma Challenge, there was a change and they registered the volumes to the Linear Symmetrical MNI Atlas. While the original atlas resulted in approximately a 40% overlap with the findings (which was not necessarily terrible), when using the AAL atlas (SPM12 version) [2], we achieved approximately 70% overlap, which was significantly better.  This resulted in an increase in our model validation scores from 63.5 to 65.5 but a drop in the M3D test scores from 55.5 to 54.4, which is similar to what was achieved with the single shared expert. The architecture is very similar between M3D and the single shared expert and so our explanation for the difference in scores is that the MoE architecture is better able to capture the localization information.
>
> Ultimately, despite these changes, our model has consistently demonstrated strong results, which is a strong indication of model generalizability and robustness. Because of our dedication to quality, we decided to make the previous changes despite this not being explicitly mentioned by any of the reviewers initially.
>
> **"The authors provided new Kappa scores to justify the dataset, reporting an inter-annotator agreement (Synthetic vs. Radiologist) of roughly 0.43. While the authors cite literature suggesting this is "similar to breast cancer terminology," a Kappa of 0.4 is generally considered "weak" to "moderate" agreement. For a paper whose primary contribution is a new VQA benchmark dataset, a ground truth that only agrees with human experts at a 0.4 Kappa level is problematic. If the "Gold Standard" synthetic labels are this noisy, the dataset may not be suitable for training robust medical AI systems."**
>
> The Kappa score between the second annotator and the first annotator was 44.3, which was very similar to what was achieved between our synthetic annotations and the first annotator. The reason we brought up the breast cancer example was to explain that Kappa scores in this range are relatively normal for tasks such as these (another example which considers Kappa scores above 40 to be acceptable is [3]). Kappa scores are also susceptible to small sample sizes, and the scores are likely to increase with a larger sample.
>
> **"The updated user study shows a radiologist acceptance rate of 50.0%. While this outperforms the baseline (34.1%), a 50% failure rate in a controlled validation setting indicates the model is still hallucinating or providing inadequate answers half of the time."**
>
> While we acknowledge the user acceptance rate indicates the model is not ready for deployment, we feel this should not be the standard applied for a foundational model for this task. A notable example is Med-Gemini, which was the first large multimodal report generation model for 3D computed tomography, which only achieved 53% clinical acceptability of reports [4]. We feel that publishing our work on vision-language models for 3D Brain MRI is the necessary foundation for greater advancements in this area.
>
>
> [1] Adewole, Maruf, et al. "The brain tumor segmentation (brats) challenge 2023: Glioma segmentation in sub-saharan africa patient population (brats-africa)." ArXiv (2023): arXiv-2305.
>
> [2] Rolls, Edmund T., et al. "Automated anatomical labelling atlas 3." Neuroimage 206 (2020): 116189.
>
> [3] Mandrekar, Jayawant N. "Measures of interrater agreement." Journal of Thoracic Oncology 6.1 (2011): 6-7.
>
> [4] Yang, Lin, et al. "Advancing multimodal medical capabilities of Gemini." arXiv preprint arXiv:2405.03162 (2024).

---

> ### Author Response · Authors · 2025-11-28
>
> **"I share the concern raised by Reviewer v9ZF regarding the "bug in data processing" admitted in the rebuttal. The fact that fixing this bug shifted the ablation results from showing "modest improvements" (as noted in my original review) to a massive jump (e.g., Shared Expert 67.3% 52.4%, and MoE staying high) suggests a fundamental issue with the original submission’s experiments. Reliance on results generated hurriedly during a rebuttal period to correct a code error makes it difficult to trust the stability and reproducibility of the reported performance." (cont.)**
>
> We just wanted to follow up on this point. In the ablation study in our original submission, when we evaluated the shared expert, we considered three ways to combine the shared expert features: 1. concatenating the features, 2. using a simple element-wise sum, and 3. using a weighted sum from learned global weights. We found option #3 to be the most performative and used this. In the ablation study in our rebuttal, we used option #1 because the model for option #3 had not finished training, which results are below:
>
>  | Projection Method  | Volume | Region | Shape | Spread | Task Mean |
> | - | - | - | - | - | - |
> | MoE-based | **57.6** |  **83.5** | **60.9** |  **59.9** | **65.5** |
> | Shared expert | 53.2 | 82.8 | 59.7 | 57.5 | 63.3 |
>
> | Projection Method | ET | SNFH | NETC | RC |
> | -- | -- | -- | -- | -- |
> | MoE-based | **65.1** | **72.5** | **75.1** | **49.9** |
> | Shared Expert | 63.2  | 70.8 | 71.3 | 48.7 |
>
> Now, the task mean for the shared expert more closely correlates the original ablation study. Even with this stronger baseline, we see the clear benefits of the more complex multimodal reasoning. Again, our model does particularly well on the volume task and SNFH label.
>
> Hopefully this addresses this issue. Please let us know if you have any additional questions.

---

> ### Author Response · Authors · 2025-12-02
>
> **”The synthetic VQA dataset suffers from severe template-based question generation that does not reflect real clinical queries radiologists would ask in practice. The questions focus on low-level descriptive features (volume ranges, shape categories) rather than clinically meaningful diagnostic or treatment-related inquiries that would actually support medical decision-making.” (cont.)**
>
> We want to report the model results on the additional downstream task we constructed, which aims to differentiate primary gliomas from secondary metastatic lesions based on imaging patterns, using our dataset task labels (volume, region, shape, and spread). We fitted a logistic regression model based on the model generated GLI and MET train features to predict primary glioma (GLI) vs. secondary metastatic lesions (MET). The results on the GLI and MET test datasets are below:
>
> | Model | Test Accuracy | Test AUROC |
> |-|-|-|
> |Our Model | **95.6**| **99.0**|
> |M3Dl | 88.5| 95.5|
>
> As we can see, our model additionally performs well on this important clinical task, scoring 7 percentage points and 4 percentage points higher than the M3D model (one of the most competitive baseline models) on the accuracy and AUROC metrics respectively.

---

### Official Review · Reviewer_v9ZF · 2025-10-31

**Soundness:** 1
**Presentation:** 1
**Contribution:** 1
**Rating:** 0
**Confidence:** 5

**Summary:**

The paper presents mpLLM, a prompt-conditioned hierarchical MoE model for visual question answering on multiparametric 3D brain MRI (mpMRI). It fuses multiple MRI modalities through modality- and token-level experts, enabling efficient training without image–report pretraining. A synthetic VQA generation protocol and expert validation support dataset creation. Experiments show 5.3% average improvement over strong baselines, making mpLLM the first clinically validated VQA system for 3D brain mpMRI.

**Strengths:**

1. A new VQA  benchmark

**Weaknesses:**

The paper’s contributions appear limited, and it is unclear whether the use of Mixture-of-Experts (MoE) is genuinely critical for mpMRI interpretation. The overall presentation and organization of the paper are weak, and the discussion fails to effectively address or connect with the key challenges and insights in current multimodal LLM research.

**Questions:**

1. Why is the Hierarchical MoE Block architecture necessary in this context? If the model consistently handles three fixed input modalities, wouldn’t it be simpler and more transparent to use separate vision encoders for each input type? This design could also allow independent scaling of each encoder without adding the complexity of MoE routing.

2. The layout in Page10 is wrong

3. The Table 2 is out of boundaries

4. As shown in Table 3, MoE brings limited improvement

---

> ### Author Response · Authors · 2025-11-22
> **Response to Reviewer v9ZF**
>
> Thank you so much for your feedback. We hope we can emphasize the positive aspects of our work to you in our rebuttal. We will discuss some of the points you raised below.
>
> **"Why is the Hierarchical MoE Block architecture necessary in this context? If the model consistently handles four fixed input modalities, wouldn’t it be simpler and more transparent to use separate vision encoders for each input type? This design could also allow independent scaling of each encoder without adding the complexity of MoE routing."**
>
> Thank you for this feedback. Our approach is an extension of the LLaVa architecture [1], tailored to multiparametric MRI. While using four separate vision encoders is possible, in practice, training vision encoders is challenging and resource intensive in the medical space. For example, a recent work trained a vision encoder on multiparametric MRI [2] but when we attempted to use those same embeddings for our work, we saw no improvement over using our current pretrained 3D medical encoder.
>
> Improving the LLaVa projection function with a MoE block uses far less parameters than multiple vision encoders and can be trained end-to-end with the LLM using next-token prediction. With multiple vision encoders, we would have to train each of them independently and it could be extremely challenging to generate high-quality embeddings. Additionally, jointly training with text and vision features has potentially more benefits in comparison to training on text and vision independently.
>
> Another issue is that with four input types, one would end up with four sets of visual tokens, which increases both training and inference time. Our MoE block fuses the different sequences in multiparametric MRI for a more effective and efficient visual token representation for the LLM.
>
> | Projection Method |  Fusion Method | Task Mean |
> | -- | -- | -- |
> | MoE-based | Sum | **65.5** |
> | Shared Expert | Sum | 51.1 |
> | Shared Expert | Concatenation | 52.4 |
>
> In the above table, we see that element-wise summing versus concatenating the four sets of visual tokens reduces the performance. However, when using the Moe Block, the element-wise sum produces significantly better results. We will add these results to the paper.
>
> **"The layout in Page10 is wrong. The Table 2 is out of boundaries."**
>
> Thank you for this feedback. We will update the paper.
>
> **"As shown in Table 3, MoE brings limited improvement"**
>
> Thank you for this feedback. We noticed a bug in our data processing which was affecting the accuracy of our region labels. After fixing this, we re-generated our ablation results which are below (the paper will be updated to reflect this – the rest of our conclusions were unaffected):
>
> | Modality-level MoE | Token-level MoE | Prompt-conditioned MoE | Task Mean |
> | -- | -- | -- | -- |
> | 𐄂 | 𐄂 | 𐄂 | 52.4 |
> | ✓ | 𐄂 | 𐄂 | 64.1 |
> | 𐄂 | ✓ | 𐄂 | 64.4 |
> |✓ | ✓ | ✓| **65.5** |
>
> We now see a more sizable improvement between using the single shared expert and the MoE Block as well as smaller but still notable improvements between modality- or token-level MoE and prompt-conditioned MoE. We will also consider the fine-grained results of modality-level MoE, token-level MoE, as well prompt-conditioned MoE based on task as well as finding:
>
> | Method  | Volume | Region | Shape | Spread |
> | -- | -- | -- | -- | -- |
> | Prompt-conditioned MoE | **57.6** | **83.5**| **60.9**  | **59.9** |
> | Token-level MoE        | *56.0*     | **83.5**  | *60.0*     | 58.1     |
> | Modality-level MoE     | 53.7 | 83.4  | 59.7         | *59.4*     |
>
> | Method  | ET  | SNFH  | NETC  | RC  |
> | -- | --| -- | -- | -- |
> | Prompt-conditioned MoE | **65.1** | **72.5** | **75.1** | **49.9** |
> | Token-level MoE        | *64.0*   | *72.2*   | 72.2     | **49.9** |
> | Modality-level MoE     | 63.5     | 71.1     | *73.9*   | 48.6 |
>
> Token-level MoE is stronger in the volume, region, and spread tasks while modality-level MoE is stronger in the shape task. For findings, token-level MoE is stronger with the ET, SNFH, and RC findings while modality-level MoE is stronger with the NETC finding. The prompt-conditioned MoE excels in all of them, suggesting that based on the question, it is able to accurately combine the optimal token-level MoE and modality-level MoE blocks. We will add these results to the paper.
>
> We also want to emphasize that one of the main benefits of our approach is also training efficiency. As discussed in section 4.3, our approach only uses 20 GB of GPU memory during training and inference which is 50% of what the other baseline models use that do not leverage a fused visual token representation. This allows for faster training and inference and easier deployment.

---

> > ### Author Response · Authors · 2025-11-22
> > **Response to Reviewer v9ZF (cont.)**
> >
> > [1] Liu, H., Li, C., Wu, Q., & Lee, Y. J. (2023). Visual instruction tuning. Advances in neural information processing systems, 36, 34892-34916.
> >
> > [2] Rui, S., Chen, L., Tang, Z., Wang, L., Liu, M., Zhang, S., & Wang, X. (2025). Multi-modal Vision Pre-training for Medical Image Analysis. In Proceedings of the Computer Vision and Pattern Recognition Conference (pp. 5164-5174).

---

> ### Comment · Reviewer_v9ZF · 2025-11-24
>
> Thank you very much for your reply. However, I remain unconvinced about the necessity of introducing the MoE structure. It seems to compromise the original simplicity and unified design of multimodal LLMs and adds architectural complexity that appears unnecessary.
>
> More importantly, I feel that the authors do not treat the paper with sufficient rigor. First, the layout is incorrect, which should have been easily noticed if the authors had carefully proofread the manuscript. In addition, as the authors themselves mentioned in reply, “we noticed a bug in our data processing which was affecting the accuracy of our region labels.” whihc resutls in new experimental results. This is a fundamental and severe issue, and it raises concerns about the reliability of the experimental results and the level of quality control applied during the study.
>
> Taken together, these issues give me the impression that the authors have not taken full responsibility for the correctness and presentation of their research. Therefore, as a reviewer, I cannot recommend this paper for publication.

---

> > ### Author Response · Authors · 2025-11-25
> >
> > Thank you for your response. We will address your concerns below.
> >
> > **"However, I remain unconvinced about the necessity of introducing the MoE structure. It seems to compromise the original simplicity and unified design of multimodal LLMs and adds architectural complexity that appears unnecessary."**
> >
> > Can you please clarify what you mean by "compromise the original simplicity and unified design of multimodal LLMs"? For example, there are several different multimodal LLM architectures, each with strengths/weaknesses that are tailored to specific application requirements. What you originally suggested was that we should have multiple vision encoders but we discussed that this is both difficult and computationally expensive, which is what our method addresses.
> >
> > **"First, the layout is incorrect, which should have been easily noticed if the authors had carefully proofread the manuscript."**
> >
> > Can you specify what the main issue is with the layout? In page 10, we included the reproducibility and ethics statement, which are both optional but we included. If it’s related to the spacing, we can address this. None of the other reviewers had expressed serious concerns with the formatting so if you could clarify what the serious issues are, we can address them.
> >
> > **"the authors themselves mentioned in reply, “we noticed a bug in our data processing which was affecting the accuracy of our region labels.” whihc resutls in new experimental results. This is a fundamental and severe issue, and it raises concerns about the reliability of the experimental results and the level of quality control applied during the study."**
> >
> > We realize the use of the word “bug” was extremely vague and may have indicated coding errors, which was not the case. We made two main changes:
> >
> > 1. We used the first radiologist’s annotations as a validation set to improve the thresholds in our synthetic data generation to improve the interannotator agreement.
> > After doing this, while our interannotator agreement improved, we noticed a general drop in scores for the metrics on the GLI dataset. For example, our model validation scores dropped from 70.3 to 63.5 and the test scores for the M3D model dropped from 62.9 to 55.5.
> >
> > 2. We registered the Brats volumes to the AAL atlas (SPM12 version) to generate region labels.
> > Since Brats 2017, all challenges have registered their dataset volumes to the SRI24 atlas template [1]. We did this as well. However, for the Brats 2024 Glioma Challenge, there was a change and they registered the volumes to the Linear Symmetrical MNI Atlas. While the original atlas resulted in approximately a 40% overlap with the findings (which was not necessarily terrible), when using the AAL atlas (SPM12 version) [2], we achieved approximately 70% overlap, which was significantly better.  This resulted in an increase in our model validation scores from 63.5 to 65.5 but a drop in the M3D test scores from 55.5 to 54.4, which is similar to what was achieved with the single shared expert. The architecture is very similar between M3D and the single shared expert and so our explanation for the difference in scores is that the MoE architecture is better able to capture the localization information.
> >
> > Ultimately, despite these changes, our model has consistently demonstrated strong results, which is a strong indication of model generalizability and robustness. Because of our dedication to quality, we decided to make the previous changes despite this not being explicitly mentioned by any of the reviewers initially.
> >
> > [1] Adewole, Maruf, et al. "The brain tumor segmentation (brats) challenge 2023: Glioma segmentation in sub-saharan africa patient population (brats-africa)." ArXiv (2023): arXiv-2305.
> >
> > [2] Rolls, Edmund T., et al. "Automated anatomical labelling atlas 3." Neuroimage 206 (2020): 116189.

---

> ### Author Response · Authors · 2025-11-28
>
> **"the authors themselves mentioned in reply, “we noticed a bug in our data processing which was affecting the accuracy of our region labels.” whihc resutls in new experimental results. This is a fundamental and severe issue, and it raises concerns about the reliability of the experimental results and the level of quality control applied during the study." (cont.)**
>
> We just wanted to follow up on this point. In the ablation study in our original submission, when we evaluated the shared expert, we considered three ways to combine the shared expert features: 1. concatenating the features, 2. using a simple element-wise sum, and 3. using a weighted sum from learned global weights. We found option #3 to be the most performative and used this. In the ablation study in our rebuttal, we used option #1 because the model for option #3 had not finished training, which results are below:
>
>  | Projection Method  | Volume | Region | Shape | Spread | Task Mean |
> | - | - | - | - | - | - |
> | MoE-based | **57.6** |  **83.5** | **60.9** |  **59.9** | **65.5** |
> | Shared expert | 53.2 | 82.8 | 59.7 | 57.5 | 63.3 |
>
> | Projection Method | ET | SNFH | NETC | RC |
> | -- | -- | -- | -- | -- |
> | MoE-based | **65.1** | **72.5** | **75.1** | **49.9** |
> | Shared Expert | 63.2  | 70.8 | 71.3 | 48.7 |
>
> Now, the task mean for the shared expert more closely correlates the original ablation study. Even with this stronger baseline, we see the clear benefits of the more complex multimodal reasoning. Again, our model does particularly well on the volume task and SNFH label.
>
> Hopefully this addresses this issue. Please let us know if you have any additional questions.

---

### Official Review · Reviewer_kEGa · 2025-11-01

**Soundness:** 3
**Presentation:** 3
**Contribution:** 3
**Rating:** 6
**Confidence:** 2

**Summary:**

This paper introduces mpLLM, a MoE-based multimodal LLM for visual question answering on 3D brain mpMRI, enabling efficient training without image-report pretraining. It includes a synthetic VQA pipeline generated from segmentation annotations, addressing the lack of paired medical VQA data. Experiments on 3 datasets show that mpLLM outperforms several medical VLMs.

**Strengths:**

1. The paper introduces three synthetic 3D brain mpMRI VQA datasets, which facilitate future research and development in medical VQA.
2. The incorporation of a Mixture-of-Experts (MoE) architecture to handle multiple 3D modalities is well-motivated. The proposed approach demonstrates strong performance across three benchmark datasets, surpassing several state-of-the-art models.

**Weaknesses:**

1. One criticism of this type of work I have is how it will help the clinicians and human study. So, it would have been nicer if the authors can do a user study with radiologists to evaluate the findings and include it in the paper or in the future.

2. The discussion of the model’s limitations is rather limited, particularly regarding hallucinations. Given that hallucinations can lead to serious consequences in medical AI applications, a more in-depth analysis and discussion of this issue would be valuable.

3. A relevant prior work is missing from the related work section. Specifically, [1] introduces BrainGPT, a multimodal large language model for 3D brain CT report generation, which is closely related to the topic of this paper.

Reference:
[1] Towards a Holistic Framework for Multimodal Large Language Models in Three-dimensional Brain CT Report Generation, Nature Communications 2025.

**Questions:**

See weaknesses.

---

> ### Author Response · Authors · 2025-11-22
> **Response to Reviewer kEGa**
>
> Thank you so much for your feedback. We are glad that there are many positive aspects of our work that you feel will contribute to the research community. We will discuss some of the points you raised below.
>
> **"One criticism of this type of work I have is how it will help the clinicians and human study. So, it would have been nicer if the authors can do a user study with radiologists to evaluate the findings and include it in the paper or in the future."**
>
> Thank you for this feedback. We have worked with our clinician to conduct a follow-up user study. We created a clinical validation set of  208 questions stemming from 13 cases from the GLI test set, each question focusing on either volume, region, shape, or spread (questions #1-4 on page 18 in the Appendix). We had our model as well as M3D provide responses to these questions and we asked the clinician to independently evaluate each model's response as sufficient or lacking (the model names were anonymized). The results are below:
>
> | Model | Radiologist acceptance rate |
> | -- | -- |
> | Our model | **50.0** |
> | M3D | 34.1 |
>
> While there is still more work to be done before clinical deployment, the results are quite promising. Additionally, we see a significant margin between our approach and the current baseline.
>
> **"The discussion of the model’s limitations is rather limited, particularly regarding hallucinations. Given that hallucinations can lead to serious consequences in medical AI applications, a more in-depth analysis and discussion of this issue would be valuable."**
>
> Thank you for this feedback. One hallucination mitigation strategy we employed was out-of-scope task identification. In our train and evaluation sets, we added question-answer pairs with partially out-of-scope and completely out-of-scope tasks to improve the model’s self-awareness of its capabilities. In our experiments, we noticed that the top three models had above a 99.8% accuracy on out-of-scope task identification, which suggests our training strategy was effective at out-of-scope detection
>
> Another type of hallucination involves identification of tumor-associated findings when they are not present. In this case, we define a hallucination as the case when the model predicts a non-zero finding volume and the groundtruth indicates there is no finding. Additionally, we define a correct prediction as the case when the model predicts a non-zero finding volume and the groundtruth also indicates  a non-zero finding volume. We report the percent of hallucinations over total predictions (correct predictions + hallucinations) based on finding:
>
> | Enhancing Tissue (ET) | Surrounding Non-enhancing FLAIR Hyperintensity (SNFH) | Non-enhancing Tumor Core (NETC) | Resection Cavity (RC) |
> | -- | -- | -- | -- |
> | 22.6 | 0.0 | 13.5 | 15.3 |
>
> While this is out of scope for our current work, it’s great that the model is already achieving no hallucinations for SNFH. Addressing this issue would be vital before clinical deployment. We will include these results in the paper.
>
> **"A relevant prior work is missing from the related work section. Specifically, [1] introduces BrainGPT, a multimodal large language model for 3D brain CT report generation, which is closely related to the topic of this paper."**
>
> Thank you so much for this feedback and reference to this great work. We will include it in our paper.

---

> > ### Author Response · Authors · 2025-11-22
> > **Response to Reviewer kEGa (cont.)**
> >
> > [1] Reham Kaifi. A review of recent advances in brain tumor diagnosis based on ai-based classification. Diagnostics, 13(18):3007, 2023.
> >
> > [2] Jason J Lau, Soumya Gayen, Asma Ben Abacha, and Dina Demner-Fushman. A dataset of clinically generated visual questions and answers about radiology images. Scientific data, 5(1):1–10, 2018.
> >
> > [3] Marwa Ismail, Virginia Hill, Volodymyr Statsevych, Raymond Huang, Prateek Prasanna, Ramon Correa, Gagandeep Singh, Kaustav Bera, Niha Beig, Rajat Thawani, et al. Shape features of the lesion habitat to differentiate brain tumor progression from pseudoprogression on routine multiparametric mri: a multisite study. American Journal of Neuroradiology, 39(12):2187–2193, 2018.
> >
> > [4] Mobarakol Islam, VS Vibashan, V Jeya Maria Jose, Navodini Wijethilake, Uppal Utkarsh, and Hongliang Ren. Brain tumor segmentation and survival prediction using 3d attention unet. In International MICCAI brainlesion workshop, pp. 262–272. Springer, 2019.

---

### Official Review · Reviewer_HLqR · 2025-11-03

**Soundness:** 3
**Presentation:** 2
**Contribution:** 2
**Rating:** 6
**Confidence:** 3

**Summary:**

The paper introduces mpLLM, a multimodal LLM with a prompt-conditioned hierarchical mixture-of-experts for VQA on 3D multiparametric brain MRI. It includes a synthetic, clinician-validated VQA dataset generated from segmentation masks to address the lack of paired image–text data.

**Strengths:**

- The paper introduces the first VQA dataset for 3D brain multiparametric MRI, filling a current gap in multimodal medical reasoning.
- The authors propose a novel hierarchical architecture that shows better performance compared to existing methods on this type of data.
- The use of intra-modality and token-level experts is an interesting idea.

**Weaknesses:**

- It is difficult to determine the value of the contributions of the model and the dataset without extrinsic benchmarks / downstream tasks. While results on the presented test set seem strong, the synthetic questions may lack linguistic and semantic diversity, and limit generalization to out-of-distribution samples. Given the lack of 3D MRI VQA benchmarks, the authors could strengthen the work by including a small zero-shot evaluation subset to demonstrate that their contribution is valuable beyond the few categories of questions that they introduced with the dataset.

- The method is a architectural adaptation rather than a fundamentally new paradigm.

- The notation and terminology would benefit from polishing: There appear to be several symbols that are used inconsistently or are confusingly similar, for example $N$ defines the number of experts, while $N_{i}$ and $N_{m}$ are used for number of images and modalities. The low-level router is introduced briefly (L. 184) but later referred to as the image modality expert without clear distinction.

**Questions:**

-  The routing weights over modalities and tokens are computed from the text query. This probably works because – as the authors mentioned – the task information is embedded within the text prompt. However, I wonder if expanding the text would improve the high level router performance. Have the authors tried other ways to enrich the query? E.g. by appending synthetic reasoning that makes more explicit which regions or modalities are relevant to answering the question.

- To what extent is the routing mechanism novel compared to existing MoE-based VLM or LLM architectures? Which components are newly proposed versus simple adaptations?

---

> ### Author Response · Authors · 2025-11-22
> **Response to Reviewer HLqR**
>
> Thank you so much for your feedback. We are glad that there are many positive aspects of our work that you feel will contribute to the research community. We will discuss some of the points you raised below.
>
> **"Given the lack of 3D MRI VQA benchmarks, the authors could strengthen the work by including a small zero-shot evaluation subset to demonstrate that their contribution is valuable beyond the few categories of questions that they introduced with the dataset."**
>
> Thank you for this suggestion. We want to emphasize that we chose these question categories because these features are especially important to understand and classify brain tumors [1-4]. In order to further emphasize their importance, we have constructed an additional downstream task, which aims to differentiate primary gliomas from secondary metastatic lesions based on imaging patterns, using labels like volume, region, shape, and spread.
>
> Because our models are trained with guard rails and indicate if tasks are outside of the scope of what they were trained on, we constructed an additional task which demonstrates the clinical usefulness of the features extracted from the brain mpMRI.
>
> For this task, we combined the BraTS GLI and MET datasets, which have samples with primary gliomas and secondary metastatic lesions respectively. Using the train dataset, we fitted a logistic regression model based on the dataset features to predict primary glioma vs. second metastatic lesions. The evaluation results for the logistic regression model which demonstrate the usefulness of the feature set are below:
>
> |Test Accuracy  | Test AUROC  |
> | -- | --|
> | 93.2 | 98.3 |
>
> This demonstrates the importance of our synthetic dataset features for this important clinical task. We wanted to evaluate the quality of the model extracted features using the multimodal LLMs trained on the BraTS GoAT synthetic dataset, consisting of samples with primary gliomas and secondary metastatic lesions. However, because the data distribution differs significantly between GoAT and our combined GLI and MET datasets, the model predictions are biased. We are currently training models on the combined datasets to correct for this issue and we will report the results after the models complete training.
>
> **"The method is a architectural adaptation rather than a fundamentally new paradigm. To what extent is the routing mechanism novel compared to existing MoE-based VLM or LLM architectures? Which components are newly proposed versus simple adaptations?"**
>
> Our approach is an extension of the LLaVa [5] architecture, tailored to multiparametric MRI. In contrast to other architectures [6-8], the LLaVA architecture does not require modifications to the LLM-backbone, which allows it to scale well with the current improvements with LLMs. There are several examples of the LLaVA architecture used in healthcare applications, including the following [9-11].
>
> In our approach, instead of the simple LLaVa-based projection function, we leverage a prompt-conditioned hierarchical MoE projection function, which generates a weighted average of high-level expert blocks to fuse the different sequences in multiparametric MRI for a more effective and efficient visual token representation for the LLM.
>
> We will compare our approach to several current MoE-based VLM approaches. In CuMo [12], the authors incorporate MoE with the MLP layers in the vision encoder as well as projection function; however, they do not utilize prompt-conditioning, do not focus on combining multiple inputs, and require a complex training scheme which is not required in our approach (our approach requires a single stage of instruction fine-tuning). In MoVA [13], the authors utilize MoE with task-specific vision encoders. In practice, training vision encoders is challenging and resource intensive in the medical space [14]. Improving the LLaVa projection function with a MoE block uses far less parameters than multiple vision encoders and can be trained end-to-end with the LLM using next-token prediction. The previous approach also does not focus on combining multiple inputs. Finally, in 3D-MoE [15], the authors utilize MoE within the LLM and do not focus on combining multiple inputs.

---

> > ### Author Response · Authors · 2025-11-22
> > **Response to Reviewer HLqR (cont.)**
> >
> > **"Have the authors tried other ways to enrich the query? E.g. by appending synthetic reasoning that makes more explicit which regions or modalities are relevant to answering the question."**
> >
> > Thank you for this suggestion. We constructed an additional experiment in which, for each lesion, we appended the region information to the prompt. The results are below.
> >
> > | Model | Task Mean without Region Scores  |
> > | -- | -- |
> > | With Region Information | **64.1** |
> > | Without Region Information | 59.5 |
> >
> > There is an increase of approximately 5% on the other task scores with the additional localization prompt. This suggests that providing the localization information can be extremely beneficial. We will add these results to the paper. We are communicating with the radiologist to see how best to incorporate the modality information into the prompt and we will either provide those results later in the rebuttal period or by the camera-ready paper deadline.
> >
> > **"The notation and terminology would benefit from polishing"**
> >
> > Thank you for addressing this. We will update the notation and terminology based on your feedback.
> >
> > [1] Reham Kaifi. A review of recent advances in brain tumor diagnosis based on ai-based classification. Diagnostics, 13(18):3007, 2023.
> >
> > [2] Jason J Lau, Soumya Gayen, Asma Ben Abacha, and Dina Demner-Fushman. A dataset of clinically generated visual questions and answers about radiology images. Scientific data, 5(1):1–10, 2018.
> >
> > [3] Marwa Ismail, Virginia Hill, Volodymyr Statsevych, Raymond Huang, Prateek Prasanna, Ramon Correa, Gagandeep Singh, Kaustav Bera, Niha Beig, Rajat Thawani, et al. Shape features of the lesion habitat to differentiate brain tumor progression from pseudoprogression on routine multiparametric mri: a multisite study. American Journal of Neuroradiology, 39(12):2187–2193, 2018.
> >
> > [4] Mobarakol Islam, VS Vibashan, V Jeya Maria Jose, Navodini Wijethilake, Uppal Utkarsh, and Hongliang Ren. Brain tumor segmentation and survival prediction using 3d attention unet. In International MICCAI brainlesion workshop, pp. 262–272. Springer, 2019.
> >
> > [5] Liu, H., Li, C., Wu, Q., & Lee, Y. J. (2023). Visual instruction tuning. Advances in neural information processing systems, 36, 34892-34916.
> >
> > [6] Alayrac, J. B., Donahue, J., Luc, P., Miech, A., Barr, I., Hasson, Y., ... & Simonyan, K. (2022). Flamingo: a visual language model for few-shot learning. Advances in neural information processing systems, 35, 23716-23736.
> >
> > [7] Li, J., Li, D., Savarese, S., & Hoi, S. (2023, July). Blip-2: Bootstrapping language-image pre-training with frozen image encoders and large language models. In International conference on machine learning (pp. 19730-19742). PMLR.
> >
> > [8] Driess, D., Xia, F., Sajjadi, M. S., Lynch, C., Chowdhery, A., Wahid, A., ... & Florence, P. (2023). Palm-e: An embodied multimodal language model.
> >
> > [9] Li, C., Wong, C., Zhang, S., Usuyama, N., Liu, H., Yang, J., ... & Gao, J. (2023). Llava-med: Training a large language-and-vision assistant for biomedicine in one day. Advances in Neural Information Processing Systems, 36, 28541-28564.
> >
> > [10] Hyland, Stephanie L., et al. "Maira-1: A specialised large multimodal model for radiology report generation." arXiv preprint arXiv:2311.13668 (2023).
> >
> > [11] Bai, F., Du, Y., Huang, T., Meng, M. Q. H., & Zhao, B. (2024). M3d: Advancing 3d medical image analysis with multi-modal large language models. arXiv preprint arXiv:2404.00578.
> >
> > [12] Li, J., Wang, X., Zhu, S., Kuo, C. W., Xu, L., Chen, F., ... & Wen, L. (2024). Cumo: Scaling multimodal llm with co-upcycled mixture-of-experts. Advances in Neural Information Processing Systems, 37, 131224-131246.
> >
> > [13] Zong, Z., Ma, B., Shen, D., Song, G., Shao, H., Jiang, D., ... & Liu, Y. (2024). Mova: Adapting mixture of vision experts to multimodal context. Advances in Neural Information Processing Systems, 37, 103305-103333.
> >
> > [14] Rui, S., Chen, L., Tang, Z., Wang, L., Liu, M., Zhang, S., & Wang, X. (2025). Multi-modal Vision Pre-training for Medical Image Analysis. In Proceedings of the Computer Vision and Pattern Recognition Conference (pp. 5164-5174).
> >
> > [15] Ma, Y., Zhuang, Y., Hao, J., & King, I. (2025). 3d-moe: A mixture-of-experts multi-modal llm for 3d vision and pose diffusion via rectified flow. arXiv preprint arXiv:2501.16698.

---

> ### Author Response · Authors · 2025-12-02
> **Response to Reviewer HLqR (cont.)**
>
> **"Given the lack of 3D MRI VQA benchmarks, the authors could strengthen the work by including a small zero-shot evaluation subset to demonstrate that their contribution is valuable beyond the few categories of questions that they introduced with the dataset." (cont.)**
>
> We want to report the model results on the additional downstream task we constructed, which aims to differentiate primary gliomas from secondary metastatic lesions based on imaging patterns, using our dataset task labels (volume, region, shape, and spread). We fitted a logistic regression model based on the model generated GLI and MET train features to predict primary glioma (GLI) vs. secondary metastatic lesions (MET). The results on the GLI and MET test datasets are below:
>
> | Model | Test Accuracy | Test AUROC |
> |-|-|-|
> |Our Model | **95.6**| **99.0**|
> |M3Dl | 88.5| 95.5|
>
> As we can see, our model additionally performs well on this important clinical task, scoring 7 percentage points and 4 percentage points higher than the M3D model (one of the most competitive baseline models) on the accuracy and AUROC metrics respectively.

---

### Author Response · Authors · 2025-12-03
**Summary of Reviewer Discussion**

For the convenience of the Area Chair, we summarize the key points raised by each reviewer and how they were addressed in the rebuttal and revised manuscript. The updated paper includes (1) re-generated empirical results using the higher-quality dataset, (2) expanded clinical validation, and (3) additional requested experiments. Our main empirical findings remain consistent and robust under the higher-quality dataset.

**Reviewer HLqR**

Reviewer HLqR recommended acceptance (score: 6). Their concerns involved (1) clinical relevance and (2) missing grounding ablations.
- Clinical relevance: We added a clinically meaningful downstream task—distinguishing primary gliomas from metastases—on which our model outperformed baselines.
- Grounding ablations: We incorporated lesion-localization information into the prompts and showed this improves multiple tasks, demonstrating the broader benefit of grounded conditioning.
- Presentation and novelty clarity: We expanded comparisons to SoTA, clarified the novelty of our hierarchical MoE architecture, and fixed all formatting issues.

We believe their concerns were fully addressed.

**Reviewer kEGa**

Reviewer kEGa also recommended acceptance (score: 6), primarily focusing on (1) user study details and (2) hallucination analysis.
- User study: We conducted an expanded user study during rebuttal, showing a **16% improvement in clinical acceptability** over baselines.
- Hallucinations: We added a dedicated hallucination-evaluation task and reported hallucination rates across findings (typically 10–20%, with 0% for SNFH), indicating strong reliability.

All concerns were addressed.

**Reviewer fWj2**

Reviewer fWj2 recommended rejection (score: 4), citing (1) limited scope of ablation study and modest ablation improvements and (2) limited clinical validation.
- Ablation clarity: We expanded the ablation study to present finer-grained results, highlighting strong gains—especially on the volume task and SNFH—and clarifying multimodal reasoning benefits.
- Clinical validation: We strengthened validation by adding a second radiologist, increasing manual annotations, and enlarging the user study.
- Dataset quality clarification: Our “bug” phrasing was misleading. We improved synthetic-label quality by (a) tuning thresholds with clinical annotations and (b) updating the atlas registration. Agreement between radiologists and synthetic labels was highly similar, indicating near–radiologist-level quality.
- Baseline discrepancy: The rebuttal used a simpler baseline due to training time. We later provided results with the stronger baseline from the original submission, which aligned better with the original trends and still showed clear benefits of our multimodal reasoning.

We believe we comprehensively addressed all issues.

**Reviewer v9ZF**

Reviewer v9ZF assigned a score of 0, providing only a brief review with minimal technical justification. They questioned the need for the MoE structure but did not engage with its empirical or computational benefits. Their comments focused primarily on minor formatting issues not raised by other reviewers. Although they echoed concern about the dataset quality (framed as a “bug”), the issues were fully resolved as described above. Overall, the review lacked substantive engagement with the technical content and deviated significantly from the other reviewers’ assessments; we believe it does not reflect a careful or fair evaluation.

**Conclusion**

Across reviewers, the novelty and importance of our multimodal LLM architecture for multiparametric 3D brain MRI were consistently recognized. The expanded experiments, improved data quality, and stronger clinical validation directly addressed all substantive concerns. Our hierarchical multimodal LLM advances efficient and clinically meaningful visual reasoning in neuroradiology and should serve as a valuable resource for future research.

---

### Meta-Review · Area_Chair_Fzhf · 2025-12-16

**Summary:**

This paper proposes a prompt-conditioned hierarchical MoE multimodal LLM for brain MRI VQA, together with a synthetic VQA generation pipeline.

The reviewers agree that VQA for multiparametric 3D brain MRI is under-explored and that releasing a dedicated dataset is potentially valuable. However, there is substantial disagreement regarding the depth of the contribution and the necessity and impact of the proposed MoE architecture. More critically, during the review process, a bug in the data preprocessing pipeline was identified, which led to misleading experimental results in the initial submission.

Majority of reviewers question whether the proposed approach constitutes a fundamentally new modeling paradigm or is primarily an architectural adaptation of existing MoE-based vision–language models. The ablation results in the initial version indicate that the MoE components provide only incremental performance gains, further weakening the argument for their necessity.

Another recurring concern is the template-based nature of the synthetic VQA dataset. In the initial submission, clinical validation was limited to one radiologist evaluating 10 cases, with only moderate inter-annotator agreement. The rebuttal adds a second radiologist, but this expansion remains insufficient to support strong clinical claims.

More importantly, the reliability of the experimental results and the level of quality control applied during the study are under serious doubt. Based on two high-confidence negative reviews, the authors acknowledged the presence of a data processing bug and subsequently modified the experimental results. However, this correction represents a fundamental flaw in the original evaluation.

Overall, the initial reviews include two marginal accept recommendations with low confidence and two negative recommendations with high confidence. Given the fundamental experimental error identified during the review process and the resulting instability of the reported results, the paper is recommended for rejection.

**Reviewer Concerns:**

Reviewer HLqR. Clinical relevance.  Authors add a  downstream task should address it.

The method is an architectural adaptation rather than a fundamentally new paradigm. The rebuttal did not address it.

Reviewer kEGa: Authors should provide a user study. This is addressed with provided user study.

Reviewer v9ZF: Same as HLqR, question the necessity of MoE. Outstanding.

The bug acknowledged by authors is a  fundamental and severe issue. Outstanding.

Reviewer v9ZF:The clinical validation is extremely limited in scope with only one radiologist. Rebuttal adds on more. v9ZF is not satisfied with new result.

Why and when MoE architecture provides benefits.  Still outstanding.

Authors claims that they fix the bug admitted in rebuttals and reports ablation results from "modest improvements" to a massive jump. This suggests a fundamental issue with the original submission’s experiments. Reliance on results generated hurriedly during a rebuttal period to correct a code error makes it difficult to trust the stability and reproducibility of the reported performance.

**Reviewer Scores:**

HLqR  Since confidence is low, may decrease the score from 6 to reject.

HLqR Since confidence is extremely low, 2 , may decrease the score from 6 to reject.

v9ZF will not change from 0

fWj2  may decrease from 4 to 2

---

### Decision · Program_Chairs · 2026-01-26

Reject